# When Does Curriculum Learning Help? A Theoretical Perspective

**Raman Arora**
Johns Hopkins University
Baltimore, MD 21218
arora@cs.jhu.edu

**Yunjuan Wang**
Johns Hopkins University
Baltimore, MD 21218
ywang509@jhu.edu

**Kaibo Zhang**
Johns Hopkins University
Baltimore, MD 21218
kzhang90@jhu.edu

## Abstract

Curriculum learning has emerged as an effective strategy to enhance the training efficiency and generalization of machine learning models. However, its theoretical underpinnings remain relatively underexplored. In this work, we develop a theoretical framework for curriculum learning based on biased regularized empirical risk minimization (RERM), identifying conditions under which curriculum learning provably improves generalization. We introduce a sufficient condition that characterizes a "good" curriculum and analyze a multi-task curriculum framework, where solving a sequence of convex tasks can facilitate better generalization. We also demonstrate how these theoretical insights translate to practical benefits when using stochastic gradient descent (SGD) as an optimization method. Beyond convex settings, we explore the utility of curriculum learning for non-convex tasks. Empirical evaluations on synthetic datasets and MNIST validate our theoretical findings and highlight the practical efficacy of curriculum-based training.

## 1 Introduction

In standard supervised learning, achieving a low generalization error often requires a large number of labeled training examples and significant computational resources. In contrast, humans can rapidly learn new concepts from only a few examples by leveraging prior knowledge. This human-like ability to relate new a new concept to the knowledge they have previously learned motivates the use of prior knowledge in a new learning problem. In paradigms such as multi-task learning [Caruana, 1997], transfer learning [Weiss et al., 2016], and meta-learning [Baxter, 2000], the assumption is that related tasks share information, allowing learners to generalize more effectively. In parameter transfer frameworks [Kuzborskij and Orabona, 2013, Pentina and Lampert, 2014], this shared structure is reflected in the assumption that tasks have similar optimal parameter vectors, enabling efficient learning through initialization and fine-tuning.

Curriculum learning [Bengio et al., 2009] draws inspiration from the structured manner in which humans acquire knowledge – starting with easier concepts and gradually progressing to more difficult ones. This paradigm proposes decomposing complex learning problems into a sequence of simpler sub-tasks ordered by increasing difficulty. The central idea is that such a learning progression can improve both optimization and generalization. Bengio et al. [2009] demonstrated how learning can benefit from gradual progression of the hardness of training data. Subsequent works extend the idea to other aspects of learning, such as increasing model capacity [Karras et al., 2017, Sinha et al., 2020, Morerio et al., 2017] and increasing task difficulty [Caubrière et al., 2019, Florensa et al., 2017, Lotter et al., 2017, Sarafianos et al., 2017, Zhang et al., 2017]. We focus on curriculum learning across tasks, where parameters are transferred from simpler tasks to more complex ones.

In contrast to traditional transfer learning, which assumes all tasks are closely related, curriculum learning introduces an ordering over tasks based on their difficulty. However, such

39th Conference on Neural Information Processing Systems (NeurIPS 2025).

an ordering does not imply that all tasks are mutually similar. In fact, strong similarity is often limited to adjacent tasks [Pentina et al., 2015]. Accordingly, we assume that only pairs of consecutive tasks are related, allowing for progressive knowledge transfer. This setup accommodates scenarios where the first and last tasks may be significantly different, as long as each intermediate step is incrementally learnable. Figure 1 illustrates this structure, where successive tasks exhibit similar loss landscapes and closely aligned minimizers.

Curriculum learning offers both optimization and statistical benefits. From an optimization perspective, continuation methods [Allgower and Georg, 2012] progressively increase problem difficulty–starting with convex, smooth objectives and transitioning to more challenging nonconvex or nonsmooth objectives–thus helping the learner avoid poor local minima. Similarly, curricula that score training samples by difficulty [Weinshall et al., 2018, Weinshall and Amir, 2020] show improved convergence when training begins on simpler examples.

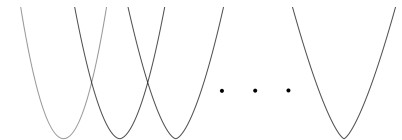

Figure 1: An illustration of potential relationship between tasks in a curriculum.

Self-paced learning [Kumar et al., 2010] adapts this idea by dynamically weighting training samples based on their inferred difficulty during training. On the statistical side, recent works [Xu and Tewari, 2022, Cohen et al., 2024] study the benefits of curriculum learning in simplified settings such as mean estimation. They show that, under appropriate conditions, learning from an easier and statistically similar source task can reduce the number of samples required to learn a target task.

In this paper, we extend previous insights to broader supervised learning problems by studying the statistical benefit of curriculum learning in the multitask setting, with a focus on the general learning setting of Vapnik [2013]. We propose a curriculum learning framework based on biased regularized empirical risk minimization (RERM)[Schölkopf et al., 2001, Denevi et al., 2019], where knowledge transfer is facilitated by incorporating a bias vector $w_0$ in the regularization term $\lambda \|w - w_0\|^2$. This inductive bias has proven effective in computer vision [Kienzle and Chellapilla, 2006, Tommasi et al., 2013], natural language processing [Daumé III, 2009], meta-learning [Pentina and Lampert, 2014, Kuzborskij and Orabona, 2017, Denevi et al., 2019, 2018], and continual learning [Li et al., 2023]. We extend this idea to design a curriculum across tasks and provide theoretical and empirical support for its effectiveness. Our key contributions are as follows.

1. We propose a biased regularization-based curriculum framework (Algorithm 1) and introduce a novel $(r, \alpha)$ condition that characterizes a 'good' curriculum. This condition is simple, natural, intuitive, and depends only on the population loss of two consecutive tasks. We show that it ensures reduced sample complexity for subsequent tasks when $r$ is small.

2. Under convexity assumptions, we provide excess risk bounds for our biased-RERM approach to curriculum learning. We show that the hardness of curriculum learning depends on the Lipschitz constants of the loss functions for each task, the local Lipschitz constants near the minimizer, the smoothness parameter of the loss function, and the quality of inductive bias obtained from learning previous tasks. These factors also determine the order of the tasks in a 'good' curriculum. We extend our analysis to efficient SGD-based training and apply our results to adversarially robust learning.

3. For nonconvex learning problems, we introduce an ERM-based curriculum learning algorithm and establish generalization guarantees via uniform convergence, showing that even in nonconvex settings, a carefully constructed curriculum can improve learning efficiency.

**Paper Organization.** In Section 2, we present the formal setup and define the $(r, \alpha)$ condition for a 'good' curriculum. Section 3 analyzes the role of biased regularization in a two-task setting. Section 4 provides theoretical guarantees for convex tasks using biased RERM and SGD. Section 5 extends our framework to nonconvex tasks using ERM. Section 6 presents empirical results on synthetic and real datasets that validate our theoretical findings.

## 2  Problem Setup

**Notation.**  Throughout, we denote scalars, vectors, and matrices with lowercase italics, lowercase bold, and uppercase bold Roman letters, respectively; e.g., $u$, $\mathbf{u}$, and $\mathbf{U}$. We use $[m]$ to denote the set $\{1, 2, \ldots, m\}$ and both $\|\cdot\|$ and $\|\cdot\|_2$ for $\ell_2$-norm. We use the standard O-notation ($\mathcal{O}$, $\Theta$ and $\Omega$).

**General Learning Problem.**  In a general learning problem, each example $\mathbf{z}$ is drawn from a data domain $\mathcal{Z}$; for instance, for standard supervised learning, $\mathcal{Z} = \mathcal{X} \times \mathcal{Y}$, where $\mathcal{X}$ is the input space and $\mathcal{Y}$ is the output space. We assume that data are drawn i.i.d. from an unknown distribution $\mathcal{D}$ over $\mathcal{Z}$. The learner has access to a training dataset $S = \{\mathbf{z}_i\}_{i=1}^n \sim \mathcal{D}^n$ consisting of $n$ i.i.d. samples. Let $\mathcal{H}$ denote the hypothesis class, where each hypothesis is parameterized by a vector $\mathbf{w} \in \mathbb{R}^m$. Let $\ell : \mathcal{Z} \times \mathcal{H} \to \mathbb{R}$ denote the loss function. The population risk with respect to the underlying population, $\mathcal{D}$, and the empirical risk on a sample $S \sim \mathcal{D}^n$ are defined, respectively, as

$$L_{\mathcal{D}}(\mathbf{w}) := \mathbb{E}_{\mathbf{z} \sim \mathcal{D}}[\ell(\mathbf{z}; \mathbf{w})], \qquad \widehat{L}_S(\mathbf{w}) := \frac{1}{|S|} \sum_{\mathbf{z} \in S} \ell(\mathbf{z}; \mathbf{w}).$$

A learning algorithm $\mathcal{A} : \mathcal{Z}^* \to \mathcal{H}$ maps any dataset $S$ to a hypothesis $\mathcal{A}(S) \in \mathcal{H}$. The goal is to design a learning algorithm with minimal *excess risk* defined as $\varepsilon(\mathbf{w}) := L_{\mathcal{D}}(\mathbf{w}) - \inf_{\mathbf{w}' \in \mathcal{H}} L_{\mathcal{D}}(\mathbf{w}')$. We consider the following classes of problems based on the structural properties of the loss function.

- **Convex:** The learning problem is *convex* if $\ell(\mathbf{z}, \mathbf{w})$ is convex in $\mathbf{w}$ for every $\mathbf{z}$.

- **Strong Convexity:** The problem is $\lambda$-*strongly convex* if $\ell(\mathbf{z}, \mathbf{w}) - \frac{\lambda}{2}\|\mathbf{w}\|_2^2$ is convex.

- **Weak Convexity:** The problem is $l$-*weakly convex* if $\ell(\mathbf{z}; \mathbf{w}) + \frac{l}{2}\|\mathbf{w}\|_2^2$ is convex in $\mathbf{w}$.

- **Lipschitz:** $\ell(\cdot, \cdot)$ is $\rho$-*Lipschitz* if, for all $\mathbf{w}_1, \mathbf{w}_2 \in \mathcal{H}$, $|\ell(\mathbf{z}; \mathbf{w}_1) - \ell(\mathbf{z}; \mathbf{w}_2)| \leq \rho \|\mathbf{w}_1 - \mathbf{w}_2\|_2$.

- **Smooth:** $\ell(\cdot, \cdot)$ is $H$-*smooth* if $\forall \mathbf{w}_1, \mathbf{w}_2 \in \mathcal{H}$, $\|\nabla_{\mathbf{w}}\ell(\mathbf{z}; \mathbf{w}_1) - \nabla_{\mathbf{w}}\ell(\mathbf{z}; \mathbf{w}_2)\|_2 \leq H \|\mathbf{w}_1 - \mathbf{w}_2\|_2$.

**Biased RERM.**  Regularized empirical risk minimization (RERM) is a popular learning algorithm known for its strong generalization performance. In its standard form, RERM returns a predictor $\mathcal{A}(S) \in \operatorname{argmin}_{\mathbf{w} \in \mathcal{H}} \widehat{L}_S(\mathbf{w}) + \frac{\mu}{2}\|\mathbf{w}\|_2^2$, where $\mu > 0$ is a regularization parameter that encourages low-norm solutions to prevent overfitting. We consider a variant called *biased* RERM, which returns

$$\mathcal{A}(S) \in \operatorname*{argmin}_{\mathbf{w} \in \mathcal{H}} \widehat{L}_S(\mathbf{w}) + \frac{\mu}{2}\|\mathbf{w} - \mathbf{w}_0\|_2^2, \tag{1}$$

where $\mathbf{w}_0 \in \mathbb{R}^m$ is a reference hypothesis that serves as an inductive bias. The regularization term now encourages solutions close to $\mathbf{w}_0$, which can be interpreted as incorporating prior knowledge into the learning. We can benefit from biased RERM if there exists a good predictor near $\mathbf{w}_0$.

**Multi-task Curriculum.**  We consider a curriculum consisting of $T$ distinct tasks. For each $t \in [T]$, the $t$-th task is defined by a specific loss function $\ell_t(\mathbf{z}; \mathbf{w})$ and an associated unknown data distribution $\mathcal{D}_t$ over the sample space $\mathcal{Z}$. Both the loss functions and data distributions may differ across tasks, capturing scenarios such as regression followed by classification, or variations in label semantics or data modalities. For each task $t \in [T]$, we draw an i.i.d. sample of size $n_t$ from the corresponding distribution, $S_t \sim \mathcal{D}_t^{n_t}$. The population and empirical risks for task $t$ are defined as

$$L_{\mathcal{D}_t}(\mathbf{w}) := \mathbb{E}_{\mathbf{z} \sim \mathcal{D}_t}[\ell_t(\mathbf{z}; \mathbf{w})], \qquad \widehat{L}_{S_t}(\mathbf{w}) := \frac{1}{n_t} \sum_{\mathbf{z} \in S_t} \ell_t(\mathbf{z}; \mathbf{w}).$$

The excess risk for task $t$ is given by $\varepsilon_t(\mathbf{w}) := L_{\mathcal{D}_t}(\mathbf{w}) - \inf_{\mathbf{w}' \in \mathcal{H}} L_{\mathcal{D}_t}(\mathbf{w}')$. The goal of curriculum learning is to learn the target task $T$ by sequentially training on all $T$ tasks, while leveraging knowledge from earlier tasks to improve generalization on the final *target* task. In this paper, we focus on curriculum learning wherein each task is solved via *biased RERM*. Specifically, we use the solution from task $t - 1$ to initialize (aka, regularize) the learning of task $t$. This is done through a bias function $\phi_t$ that maps the learned hypothesis $\widehat{\mathbf{w}}_{t-1}$ from the previous task to a bias vector for the current task. For simplicity, we assume $\phi_t$ is the identity map, i.e., the bias for task $t$ is directly given by $\widehat{\mathbf{w}}_{t-1}$. However, our framework naturally extends to more general settings where each task $t$ may have its own hypothesis class $\mathcal{H}_t$, and the bias function $\phi_t : \mathcal{H}_{t-1} \to \mathcal{H}_t$ bridges the learned hypothesis from task $t - 1$ to a suitable inductive bias for task $t$. This sequential procedure using biased RERM across tasks is formalized in Algorithm 1.

---
**Algorithm 1** Biased Regularization-based Curriculum Learning
---
**Input:** $w_0, S_1, \ldots, S_T, \mu_1, \ldots, \mu_T > 0$.
$\quad \widehat{w}_0 = w_0$.
$\quad$ **for** $t = 1, 2, ..., T$ **do**
$\quad\quad \widehat{w}_t \in \underset{w}{\text{argmin}} \left( \widehat{L}_{S_t}(w) + \frac{\mu_t}{2} \| w - \widehat{w}_{t-1} \|_2^2 \right)$.
$\quad$ **end for**
$\quad$ return: $\widehat{w}_T$.
---

$(r, \alpha)$ **Condition of the Curriculum.** To effectively apply biased RERM to task $t$, we require a good bias $\phi_t(\widehat{w}_{t-1})$, i.e., a previous solution close to a good predictor for the current task after mapped onto $\mathcal{H}_t$. Since $\widehat{w}_{t-1}$ is obtained by learning task $t - 1$, this requires that consecutive tasks be similar enough for the prior solution to be informative. We formalize this similarity using the $(r_t, \alpha_t)$ *condition*, which relates the excess risks of two consecutive tasks. Specifically, we assume that

$$\inf_{w': \| w' - \phi_t(w) \|_2 \leq r_t} \varepsilon_t(w') \leq \alpha_t \varepsilon_{t-1}(w), \tag{2}$$

for some constants $r_t > 0$ and $\alpha_t \in (0, 1)$. When this condition holds, we say that tasks $t - 1$ and $t$ satisfy the $(r_t, \alpha_t)$ *condition*. We will assume $\phi_t$ is the identity mapping only for simplicity in our core theorems.

Intuitively, this condition means that if a predictor w has small excess risk on task $t - 1$, then there exists a predictor $w'$ within a ball of radius $r_t$ centered at w, with excess risk at most $\alpha_t \epsilon$ on task $t$. Thus, a solution to task $t - 1$ can serve as a useful initialization or inductive bias for task $t$.

We note that since the scale of the loss functions is arbitrary, one can always apply affine rescaling to them so that $\alpha_t$ is the same across all tasks. Therefore, for simplicity, we assume $\alpha_t = \alpha \in (0, 1) \ \forall t$. In practice, we do not rescale losses across tasks. But for theoretical clarity, assuming a constant $\alpha < 1$ allows us to streamline the presentation of our results without loss of generality. If needed, our theorems could be extended to carry task-dependent $\alpha_t$ values throughout. Further, we emphasize that we do not need condition (2) to hold for all $w \in \mathbb{R}^m$. Since we initialize the learning for task $t$ with a predictor that generalizes well on task $t - 1$, it suffices if (2) holds for w with $\epsilon_{t-1}(w) \leq \epsilon$ for sufficiently small $\epsilon \in (0, 1)$. However, we may need to set $\epsilon$ differently for different settings. So, for convenience, and without loss of generality, we state the condition as in (2).

## 3 Warm-up: Curriculum Learning with Two Tasks

In this section, we illustrate the role of biased RERM in curriculum learning by analyzing a simple setting with only two tasks, i.e., $T = 2$. The goal is to learn the second (target) task by first learning the first (source) task. Let $w_1^\star \in \text{argmin}_w L_{\mathcal{D}_1}(w)$, $w_2^\star \in \text{argmin}_w L_{\mathcal{D}_2}(w)$ denote the optimal predictors for the two tasks. We assume that the first task is $\lambda$-strongly convex with gradients uniformly bounded at the optimum: $\| \nabla_w \ell_1(z; w_1^\star) \|_2 \leq \rho_1, \forall z \in \mathcal{Z}$. Second task is $\rho_2$-Lipschitz.

The curriculum solves the first task using empirical risk minimization (ERM), and the second task using biased regularized ERM. This procedure is described in Algorithm 2.

---
**Algorithm 2** Warm-up: A Two-task Curriculum
---
**Input:** $S_1, S_2, \mu_2 > 0$.
$\quad \widehat{w}_1 = \text{argmin}_w \widehat{L}_{S_1}(w)$.
$\quad \widehat{w}_2 = \text{argmin}_w \left( \widehat{L}_{S_2}(w) + \frac{\mu_2}{2} \| w - \widehat{w}_1 \|_2^2 \right)$.
$\quad$ return: $\widehat{w}_2$.
---

**Theorem 3.1.** If the second task is convex, then setting $\mu_2 = \frac{2\rho_2}{\left( \| w_2^\star - w_1^\star \|_2 + \frac{\rho_1}{\lambda \sqrt{n_1}} \right) \sqrt{n_2}}$, we have

$$\mathbb{E}\left[\varepsilon_2(\widehat{w}_2)\right] \leq \frac{2\rho_2}{\sqrt{n_2}} \left( \| w_2^\star - w_1^\star \|_2 + \frac{\rho_1}{\lambda \sqrt{n_1}} \right).$$

**Theorem 3.2.** If the second task is $l$-weakly convex, then setting $\mu_2 = l + \frac{2\rho_2}{\left(\|\mathbf{w}_2^\star - \mathbf{w}_1^\star\|_2 + \frac{\rho_1}{\lambda\sqrt{n_1}}\right)\sqrt{n_2}}$,

we have $\mathbb{E}\left[\varepsilon_2(\widehat{\mathbf{w}}_2)\right] \leq \frac{2\rho_2}{\sqrt{n_2}}\left(\|\mathbf{w}_2^\star - \mathbf{w}_1^\star\|_2 + \frac{\rho_1}{\lambda\sqrt{n_1}}\right) + \frac{l}{2}\left(\|\mathbf{w}_2^\star - \mathbf{w}_1^\star\|_2 + \frac{\rho_1}{\lambda\sqrt{n_1}}\right)^2$.

Theorems 3.1 and 3.2 show that curriculum learning can achieve a fast generalization rate for the target task under a mild similarity assumption—specifically, when the optimal solutions of the two tasks, $\mathbf{w}_1^\star$ and $\mathbf{w}_2^\star$, are close. This proximity ensures that a hypothesis learned from the simpler first task (i.e., strongly convex) can serve as an effective bias for the more challenging second task. Importantly, the excess risk bound for the second task reflects this structure: it improves as the distance between $\mathbf{w}_1^\star$ and $\mathbf{w}_2^\star$ decreases and achieves a fast rate as the first task is easier to learn.

The regularization parameter $\mu_2$ in both theorems is chosen to optimize the theoretical bound and depends on unknown problem-specific quantities. These values are thus not intended for practical implementation. In practice, $\mu_2$ should be treated as a tunable hyperparameter, selected via validation or cross-validation. Nonetheless, the analysis reveals that a two-phase curriculum strategy—first solving a well-behaved source task, then regularizing toward its solution can yield statistically significant gains in sample efficiency for the target task.

# 4 Curriculum Learning with Multiple Convex Learning Tasks

In this section, we consider a curriculum comprising $T$ convex learning tasks that are learned sequentially. Our goal is to demonstrate the role of the $(r, \alpha)$ condition in facilitating efficient learning of the target (i.e., the $T^{\text{th}}$) task. Here, we focus on convex learning problems (with additional structure, e.g., Lipschitzness, smoothness, and non-negativity). We first provide theoretical guarantees for biased RERM under this setup (Section 4.1), then extend the analysis to computationally efficient variants such as SGD (Section 4.2) and settings where tighter bounds can be obtained by leveraging local geometry (Section 4.3). We relax the convexity assumption in Section 5.

## 4.1 Learning Convex Lipschitz Tasks using Biased RERM

We assume that each task $t \in [T]$ in the curriculum is a convex learning problem with a $\rho_t$-Lipschitz loss function. Furthermore, we assume that every pair of consecutive tasks $(t-1, t)$ satisfies the $(r_t, \alpha)$ condition for some constants $r_t > 0$ and $\alpha \in (0, 1)$. We use the biased RERM algorithm described in Algorithm 1 for curriculum learning. To highlight the benefit of the curriculum, we begin by analyzing two consecutive tasks: task $t-1$ and task $t$.

**Theorem 4.1.** Suppose task $t$ is convex and $\rho_t$-Lipschitz, and the $(r_t, \alpha)$ condition holds between tasks $(t-1, t)$. Then, setting $\mu_t = \frac{2\rho_t}{r_t\sqrt{n_t}}$ yields the following excess risk bound:

$$\mathbb{E}\left[\varepsilon_t(\widehat{\mathbf{w}}_t)\right] \leq \frac{2r_t\rho_t}{\sqrt{n_t}} + \alpha\mathbb{E}\left[\varepsilon_{t-1}(\widehat{\mathbf{w}}_{t-1})\right].$$

The result above shows that if $r_t$ is a small constant, then using $\widehat{\mathbf{w}}_{t-1}$ as the bias in biased RERM leads to a smaller sample complexity for learning task $t$. A natural setting where this occurs is when the minimizers of successive tasks are close. For example, in large language models (LLMs), task $t$–1 can represent a pretraining phase that yields a model $\widehat{\mathbf{w}}_{t-1}$ close to the minimizer of many related downstream tasks. If task $t$ is such a downstream task and its minimizer is close to that of task $t-1$, then a small perturbation of $\widehat{\mathbf{w}}_{t-1}$ yields a good predictor for task $t$. In this case, a small value of $r_t$ is justified, and the sample complexity required to generalize on task $t$ is correspondingly small.

**Proof Sketch.** To upper bound the excess risk $\varepsilon_t(\widehat{\mathbf{w}}_t)$, we begin by decomposing it as follows:

$$\mathbb{E}_{S_t}\left[\varepsilon_t(\widehat{\mathbf{w}}_t)\right] = \mathbb{E}_{S_t}\left[L_{\mathcal{D}_t}(\widehat{\mathbf{w}}_t)\right] - \inf_{\mathbf{w}} L_{\mathcal{D}_t}(\mathbf{w})$$

$$= \mathbb{E}_{S_t}\left[L_{\mathcal{D}_t}(\widehat{\mathbf{w}}_t) - \widehat{L}_{S_t}(\widehat{\mathbf{w}}_t)\right] + \mathbb{E}_{S_t}\left[\widehat{L}_{S_t}(\widehat{\mathbf{w}}_t) - \widehat{L}_{S_t}(\mathbf{w}')\right] + \left[L_{\mathcal{D}_t}(\mathbf{w}') - \inf_{\mathbf{w}} L_{\mathcal{D}_t}(\mathbf{w})\right] \quad (3)$$

where $\mathbf{w}'$ is any hypothesis independent with $S_t$. By the $(r_t, \alpha)$ condition, there exists $\mathbf{w}'$, s.t. $\|\mathbf{w}' - \widehat{\mathbf{w}}_{t-1}\|_2 \leq r_t$ and $\varepsilon_t(\mathbf{w}') \leq \alpha\varepsilon_{t-1}(\widehat{\mathbf{w}}_{t-1})$ hold. Hence, the third term in the decomposition can be

upper bounded by $L_{\mathcal{D}_t}(\mathbf{w}') - \inf_{\mathbf{w}} L_{\mathcal{D}_t}(\mathbf{w}) \leq \alpha \varepsilon_{t-1}(\widehat{\mathbf{w}}_{t-1})$. Next, consider the second term. Using the definition of biased RERM, we have:

$$
\begin{aligned}
\mathbb{E}_{S_t}\left[\widehat{L}_{S_t}(\widehat{\mathbf{w}}_t) - \widehat{L}_{S_t}(\mathbf{w}')\right] &\leq \mathbb{E}_{S_t}\left[\widehat{L}_{S_t}(\widehat{\mathbf{w}}_t) + \frac{\mu_t}{2}\|\widehat{\mathbf{w}}_t - \widehat{\mathbf{w}}_{t-1}\|_2^2 - \widehat{L}_{S_t}(\mathbf{w}')\right] \\
&\leq \mathbb{E}_{S_t}\left[\widehat{L}_{S_t}(\mathbf{w}') + \frac{\mu_t}{2}\|\mathbf{w}' - \widehat{\mathbf{w}}_{t-1}\|_2^2 - \widehat{L}_{S_t}(\mathbf{w}')\right] \leq \frac{\mu_t r_t^2}{2}.
\end{aligned}
$$

where the second inequality follows from the optimality of $\widehat{\mathbf{w}}_t$ under biased RERM and the final inequality uses the assumption $\|\mathbf{w}' - \widehat{\mathbf{w}}_{t-1}\|_2 \leq r_t$.

Finally, we consider the first term in the decomposition: $\mathbb{E}_{S_t}\left[L_{\mathcal{D}_t}(\widehat{\mathbf{w}}_t) - \widehat{L}_{S_t}(\widehat{\mathbf{w}}_t)\right]$ – the generalization gap. Using the uniform stability results from Shalev-Shwartz and Ben-David [2014], and noting that the loss function is convex and $\rho_t$-Lipschitz, the generalization gap can be bounded by $\mathbb{E}_{S_t} L_{\mathcal{D}_t}(\widehat{\mathbf{w}}_t) \leq \mathbb{E}_{S_t} \widehat{L}_{S_t}(\widehat{\mathbf{w}}_t) + \frac{2\rho_t^2}{\mu_t n_t}$. Putting the three terms together and optimizing the bound w.r.t. $\mu_t$ to minimize the sum of the first two terms yields the upper bound stated in Theorem 4.1. $\blacksquare$

**Corollary 4.2.** Assume the first task is learned with excess risk $\mathbb{E}\left[\varepsilon_1(\widehat{\mathbf{w}}_1)\right] \leq \epsilon$. Set the regularization parameter to $\mu_t = \frac{2\rho_t}{r_t\sqrt{n_t}}$, and suppose the sample size $n_t \geq \frac{4r_t^2\rho_t^2}{(1-\alpha)^2\epsilon^2}$. Then, for all tasks $t$, the excess risk is bounded as $\mathbb{E}\left[\varepsilon_t(\widehat{\mathbf{w}}_t)\right] \leq \epsilon$.

In the above Corollary 4.2, we assume that the first task is sufficiently easy to learn to a small excess risk. This can be achieved, for example, by choosing a strongly convex learning problem, using a large number of samples, or initializing from a high-quality pretrained model.

However, requiring $\epsilon$-suboptimality for all tasks may be unnecessarily strict, especially when our goal is only to achieve small excess risk on the final target task. Instead, Theorem 4.1 allows us to ensure that the excess risk forms a decreasing sequence across tasks, culminating in a final bound of $\epsilon$ only for the target task $T$. This motivates the next corollary.

**Corollary 4.3.** Suppose the first task is learned to excess risk $\mathbb{E}\left[\varepsilon_1(\widehat{\mathbf{w}}_1)\right] \leq \epsilon_1$. Set the regularization parameter as $\mu_t = \frac{2\rho_t}{r_t\sqrt{n_t}}$, and assume the sample size satisfies $n_t \geq \left(\frac{4r_t\rho_t}{(1-\alpha)\epsilon_1\left(\frac{\alpha+1}{2}\right)^{t-2}}\right)^2$. Then, for every task $t$, we have $\mathbb{E}\left[\varepsilon_t(\widehat{\mathbf{w}}_t)\right] \leq \epsilon_1\left(\frac{\alpha+1}{2}\right)^{t-1}$.

Since $\alpha < 1$, the bound $\epsilon_1\left(\frac{\alpha+1}{2}\right)^{t-1}$ decreases with $t$. This decay allows smaller sample complexity for earlier tasks in the curriculum and, correspondingly, the use of larger regularization parameters $\mu_t$. Larger $\mu_t$ yields strongly convex objectives with larger strong convexity parameters and thereby improving the computational efficiency of learning.

## 4.2 Learning Lipschitz Convex Losses with SGD

We show that, instead of using biased RERM, one can apply stochastic gradient descent (SGD) with a carefully chosen learning rate to achieve the same excess risk bound as in Theorem 4.1. The SGD procedure for task $t$ is described in Algorithm 3.

---

**Algorithm 3** SGD for task $t$

---

**Input:** $\mathbf{w}_0 = \widehat{\mathbf{w}}_{t-1}$, $S_t = \{\mathbf{z}_1, \ldots, \mathbf{z}_{n_t}\}$, $\eta_t > 0$.
    **for** $k = 1, 2, \ldots, n_t$ **do**
        $\mathbf{w}_k = \mathbf{w}_{k-1} - \eta_t \nabla_{\mathbf{w}} \ell_t(\mathbf{w}_{k-1}; \mathbf{z}_k)$.
    **end for**
    return: $\widehat{\mathbf{w}}_t = \frac{1}{n_t} \sum_{k=0}^{n_t-1} \mathbf{w}_k$.

---

For simplicity, we analyze the case of two consecutive tasks, $t-1$ and $t$, as in Section 4.1.

**Theorem 4.4.** Suppose task $t$ has a $\rho_t$-Lipschitz convex loss function and satisfies the $(r_t, \alpha)$ condition with task $t-1$. Choosing the learning rate $\eta_t = \frac{r_t}{\rho_t\sqrt{n_t}}$, the excess risk of SGD satisfies

$$
\mathbb{E}\left[\varepsilon_t(\widehat{\mathbf{w}}_t)\right] \leq \frac{r_t\rho_t}{\sqrt{n_t}} + \alpha \mathbb{E}\left[\varepsilon_{t-1}(\widehat{\mathbf{w}}_{t-1})\right].
$$

The bound above matches the result in Theorem 4.1, and thus all subsequent corollaries carry over to this setting. Crucially, the use of SGD offers computational advantages: it is an efficient single-pass algorithm and updates the model using only one example at a time. This makes it particularly appealing in large-scale or streaming settings, while still benefiting from the curriculum structure.

## 4.3 A Tighter Bound via Leveraging the Local Lipschitz Constant

To obtain a sharper excess risk bound, we refine our analysis to leverage local Lipschitz constant around the minimizers. Specifically, we define a local Lipschitz constant $\bar{\rho}_t$ over the set of predictors w with excess risk at most $\bar{\varepsilon}_t$. The intuition is that since the final hypothesis $\widehat{w}_t$ is expected to achieve small excess risk, it may suffice to control the gradient magnitude only in this restricted region–leading to a potentially smaller constant $\bar{\rho}_t \ll \rho_t$. Formally,

$$\bar{\rho}_t \geq \sup_z \sup_{w:\varepsilon_t(w)\leq\bar{\varepsilon}_t} \left\| \frac{\partial \ell_t(z;w)}{\partial w} \right\|_2. \tag{4}$$

**Theorem 4.5.** Choosing $\mu_t$ appropriately, the excess risk of curriculum learning satisfies

$$\mathbb{E}\left[\varepsilon_t(\widehat{w}_t)\right] \leq \frac{2r_t}{\sqrt{n_t}} \left( \bar{\rho}_t + \frac{6r_t(\rho_t^2 - \bar{\rho}_t^2)}{(1-\alpha)\bar{\varepsilon}_t\sqrt{n_t}} \right) + \frac{1+\alpha}{2} \mathbb{E}\left[\varepsilon_{t-1}(\widehat{w}_{t-1})\right].$$

We note that Theorem 4.5 recovers Theorem 4.1 as a special case by setting $\bar{\rho}_t = \rho_t$. However, a meaningful improvement can be obtained when $n_t$ is large and $\bar{\rho}_t \ll \rho_t$. In such cases, the upper bound $\approx \frac{2r_t}{\sqrt{n_t}}\bar{\rho}_t + \frac{1+\alpha}{2}\mathbb{E}\left[\varepsilon_{t-1}(\widehat{w}_{t-1})\right]$. Moreover, the bound depends on both $\bar{\varepsilon}_t$ and $\bar{\rho}_t$. Since $\bar{\rho}_t = \bar{\rho}_t(\bar{\varepsilon}_t)$ can be interpreted as a non-decreasing function of $\bar{\varepsilon}_t$ (by definition in (4)), one can minimize the overall upper bound by balancing the two terms: $\bar{\rho}_t$ and $\frac{6r_t(\rho_t^2-\bar{\rho}_t^2)}{(1-\alpha)\bar{\varepsilon}_t\sqrt{n_t}}$. This offers an additional degree of flexibility in tightening the excess risk bound.

## 4.4 Learning Smooth and Nonnegative Convex Losses with Biased RERM

In this section, we assume that the loss functions satisfy smoothness, rather than Lipschitz continuity. Specifically, we assume that for all z, the loss function $\ell_t(z;w)$ of task $t$ is convex, nonnegative, and $H_t$-smooth with respect to $w \in \mathbb{R}^m$. Moreover, tasks $t-1$ and $t$ satisfy the $(r_t, \alpha)$ condition for constants $r_t > 0$ and $\alpha \in (0,1)$. Let $L_t^\star = \inf_w L_{\mathcal{D}_t}(w)$. As in earlier sections, we employ biased RERM to learn each task and focus our analysis on two consecutive tasks.

**Theorem 4.6.** Setting the regularization parameter $\mu_t = \max\{\frac{(2+6\alpha)H_t}{(1-\alpha)n_t}, \frac{1}{r_t}\sqrt{\frac{32H_tL_t^\star}{n_t}}\}$, we have

$$\mathbb{E}\left[\varepsilon_t(\widehat{w}_t)\right] \leq \sqrt{\frac{32L_t^\star H_t r_t^2}{n_t}} + \frac{9H_t r_t^2}{(1-\alpha)n_t} + \frac{1+\alpha}{2}\mathbb{E}\left[\varepsilon_{t-1}(\widehat{w}_{t-1})\right].$$

The proof closely mirrors the argument used in Theorem 4.1, relying on the same excess risk decomposition from equation (3). The second and third terms in the decomposition are bounded using the same techniques as before. For the first term–the generalization gap–we apply a stability-based argument for smooth, nonnegative losses. Specifically, from standard results on uniform stability for smooth objectives, we obtain $\mathbb{E}_{S_t}L_{\mathcal{D}_t}(\widehat{w}_t) \leq \left(\frac{\mu_t n_t + H_t}{\mu_t n_t - H_t}\right)^2 \mathbb{E}_{S_t}\widehat{L}_{S_t}(\widehat{w}_t)$ as long as $\mu_t n_t > H_t$.

Theorem 4.6 provides an *optimistic rate* for smooth convex losses. In the realizable case where $L_t^\star = 0$, we obtain a fast rate of $\mathcal{O}(1/n_t)$. Note that for this result to hold, the loss function must be well-defined over the entire domain $w \in \mathbb{R}^m$. Similar to Theorem 4.1, the benefit of the curriculum becomes evident when each $r_t$ is small, enabling significant gains in sample efficiency. Even in the absence of a curriculum, this analysis yields an optimistic bound by replacing $r_t$ with a larger constant. Thus, incorporating curriculum learning never worsens the sample complexity (up to the parameters $r_t$ and $\alpha$), and often leads to notable improvements.

While our analysis thus far has focused on multi-task curricula, the framework naturally extends to the single-task setting. Suppose we are given a single learning task and aim to construct an effective curriculum within its dataset. One strategy is to begin training on a subset of "easy" examples–those

for which the loss is small–and then gradually incorporate the full training distribution. This aligns with the original motivation behind curriculum learning [Bengio et al., 2009], where the learner is first exposed to simpler examples and then to increasingly complex ones.

From Theorem 4.6, the excess risk bound depends on the regularization radius $r$, transferability parameter $\alpha$, smoothness $H_t$, and the optimal population loss $L_t^\star$. We therefore aim to identify a subset of training examples that satisfies two goals: (1) the resulting subtask is *similar* to the original task in the sense that the pair satisfies a $(r, \alpha)$ condition with small $r$ and $\alpha$, and (2) the subtask has a smaller optimal risk $L_t^\star$, thereby reducing the sample complexity required to learn it.

Practically, this involves selecting a 'good' subset of the training data–i.e., a collection of examples with low loss values under an initial model–to define an auxiliary task. The learner can then solve this easier task first and use the resulting solution as a bias to efficiently solve the full task. This strategy mirrors the continuation principle embedded in curriculum learning: leveraging simple concepts as stepping stones to learn more complex ones. This idea is confirmed by Saglietti et al. [2022] and Abbe et al. [2023]. They considered specific settings and selected sparse data and low noise data as the 'good' subset.

## 5 Curriculum Learning without Convexity

Deep learning has become the cornerstone of recent advances in artificial intelligence and machine learning, powering state-of-the-art performance across domains such as vision, language, and robotics. At the heart of deep learning is the training of deep neural networks–an inherently nonconvex optimization problem. In this section, we investigate the benefits of curriculum learning in this nonconvex setting, focusing on tasks whose loss functions are nonconvex but Lipschitz continuous.

We assume a curriculum composed of $T$ tasks, where each task $t$ has a $\rho_t$-Lipschitz, nonconvex loss function. As before, we assume each pair of consecutive tasks satisfies the $(r_t, \alpha)$ condition for some $r_t > 0$, $\alpha \in (0, 1)$. Unlike the convex case, where we use biased RERM, we propose an ERM-based strategy for nonconvex problems. For each task $t$, we select a solution by minimizing empirical loss over a ball of radius $r_t$ centered at $\widehat{w}_{t-1}$. This is formalized in Algorithm 4.

---

**Algorithm 4** ERM-based Curriculum Learning

**Input:** $w_0, S_1, \ldots, S_T, r_1, \ldots, r_T > 0$.
   $\widehat{w}_0 = w_0$.
   **for** $t = 1, 2, ..., T$ **do**
      $\widehat{w}_t \in \underset{w:\|w-\widehat{w}_{t-1}\|_2 \leq r_t}{\operatorname{argmin}} \widehat{L}_{S_t}(w)$.
   **end for**
   return: $\widehat{w}_T$.

---

When $\widehat{L}_{S_t}(w)$ is convex, the projection-based ERM in Algorithm 4 is equivalent to biased RERM with quadratic regularization as in Algorithm 1. However, in the nonconvex case, Algorithm 4 enables a broader exploration of the parameter space. Although this procedure may not be computationally efficient, in practice it can be approximated using methods such as SGD. We also note that the radius $r_t$ is used primarily for theoretical analysis; in practice, it can be treated as a tunable parameter. For example, early stopping can serve as a proxy for tuning $r_t$, by controlling how long we train on easier data subsets. Solving constrained ERM exactly is not practical in large-scale deep learning. However, the purpose of our non-convex analysis is to provide generalization guarantees for implicit approximations to this problem, such as those computed via SGD and backpropagation. From this perspective, theoretical analysis of constrained ERM remains meaningful. Next, we present a key result for this setting.

**Lemma 5.1.** Let $\delta \in (0, 1)$ and $\epsilon > 0$. If $n_t \geq \frac{8r_t^2\rho_t^2}{\epsilon^2}\left(\ln\left(\frac{2}{\delta}\right) + m\ln\left(\frac{8r_t\rho_t}{\epsilon} + 1\right)\right)$, then with probability at least $1 - \delta$ over the randomness of $S_t$,

$$\sup_{w:\|w-\widehat{w}_{t-1}\|_2 \leq r_t} |\widehat{L}_{S_t}(w) - L_{\mathcal{D}_t}(w) - \widehat{L}_{S_t}(\widehat{w}_{t-1}) + L_{\mathcal{D}_t}(\widehat{w}_{t-1})| \leq \epsilon.$$

This lemma establishes uniform concentration over $\{\ell(\mathrm{z};\mathrm{w}) - \ell(\mathrm{z};\widehat{\mathrm{w}}_{t-1}) \,\big|\, \|\mathrm{w}-\widehat{\mathrm{w}}_{t-1}\|_2 \leq r_t\}$—the loss class of shifted loss functions rather than $\{\ell(\mathrm{z};\mathrm{w}) \,\big|\, \|\mathrm{w}-\widehat{\mathrm{w}}_{t-1}\|_2 \leq r_t\}$. This avoids dependence on potentially large loss values and instead leverages the Lipschitz condition: $|\ell(\mathrm{z};\mathrm{w}) - \ell(\mathrm{z};\widehat{\mathrm{w}}_{t-1})| \leq \rho_t r_t$, which is small when $r_t$ is small.

We also remark that we can give a tighter bound and remove the log term $\ln\left(\frac{8r_t\rho_t}{\epsilon} + 1\right)$ via chaining. This can also be applied to Theorem 5.2 and Corollary 5.3 below. We have an in expectation bound

$$\mathbb{E}_{S_t}\left[\sup_{\mathrm{w}:\|\mathrm{w}-\widehat{\mathrm{w}}_{t-1}\|_2 \leq r_t} |\widehat{L}_{S_t}(\mathrm{w}) - L_{\mathcal{D}_t}(\mathrm{w}) - \widehat{L}_{S_t}(\widehat{\mathrm{w}}_{t-1}) + L_{\mathcal{D}_t}(\widehat{\mathrm{w}}_{t-1})|\right] \leq 2r_t\rho_t\sqrt{\frac{3+9m}{n_t}}.$$

The high probability bound can be derived from this using McDiarmid's Inequality.

**Theorem 5.2.** For any $\epsilon > 0$, if $n_t \geq \frac{8r_t^2\rho_t^2}{\epsilon^2}\left(\ln\left(\frac{2}{\delta}\right) + m\ln\left(\frac{8r_t\rho_t}{\epsilon} + 1\right)\right)$, then with probability at least $1-\delta$ over the randomness of $S_t$, we have $\varepsilon_t(\widehat{\mathrm{w}}_t) \leq 2\epsilon + \alpha\varepsilon_{t-1}(\widehat{\mathrm{w}}_{t-1})$.

To prove Theorem 5.2, we decompose the excess risk a bit differently from Equation (3):

$$\varepsilon_t(\widehat{\mathrm{w}}_t) = L_{\mathcal{D}_t}(\widehat{\mathrm{w}}_t) - \inf_{\mathrm{w}} L_{\mathcal{D}_t}(\mathrm{w})$$
$$= \left[L_{\mathcal{D}_t}(\widehat{\mathrm{w}}_t) - \widehat{L}_{S_t}(\widehat{\mathrm{w}}_t) - L_{\mathcal{D}_t}(\widehat{\mathrm{w}}_{t-1}) + \widehat{L}_{S_t}(\widehat{\mathrm{w}}_{t-1})\right] + \left[\widehat{L}_{S_t}(\widehat{\mathrm{w}}_t) - \widehat{L}_{S_t}(\mathrm{w}')\right]$$
$$+ \left[\widehat{L}_{S_t}(\mathrm{w}') - L_{\mathcal{D}_t}(\mathrm{w}') - \widehat{L}_{S_t}(\widehat{\mathrm{w}}_{t-1}) + L_{\mathcal{D}_t}(\widehat{\mathrm{w}}_{t-1})\right] + \left[L_{\mathcal{D}_t}(\mathrm{w}') - \inf_{\mathrm{w}} L_{\mathcal{D}_t}(\mathrm{w})\right],$$

where $\mathrm{w}'$ satisfies $\|\mathrm{w}' - \widehat{\mathrm{w}}_{t-1}\|_2 \leq r_t$ and $\varepsilon_t(\mathrm{w}') \leq \alpha\varepsilon_{t-1}(\widehat{\mathrm{w}}_{t-1})$. The fourth term is bounded by $\alpha\varepsilon_{t-1}(\widehat{\mathrm{w}}_{t-1})$ by the $(r_t, \alpha)$ condition. The second term is nonpositive as $\widehat{\mathrm{w}}_t$ is the ERM solution. The first and the third terms are each bounded by $\epsilon$ using Lemma 5.1, completing the proof.

As in the convex case, smaller values of $r_t$ lead to lower sample complexity requirements. We conclude with a high-probability bound for the entire curriculum:

**Corollary 5.3.** Assume $\varepsilon_1(\widehat{\mathrm{w}}_1) \leq \epsilon$. If $n_t \geq \frac{32r_t^2\rho_t^2}{(1-\alpha)^2\epsilon^2}\left(\ln\left(\frac{2T}{\delta}\right) + m\ln\left(\frac{16r_t\rho_t}{(1-\alpha)\epsilon} + 1\right)\right)$, for all $t \in 2,\ldots,T$, then Algorithm 4 ensures that with probability at least $1-\delta$, we have that $\varepsilon_T(\widehat{\mathrm{w}}_T) \leq \epsilon$.

# 6 Experiments

We conduct a simple empirical study using both synthetic and real dataset to support our theory. First, we investigate whether curriculum learning can enhance large-margin classifiers on separable data by first training on easy examples and then fine-tuning on harder ones. Specifically, we construct a binary classification task using mixtures of two-centered Gaussians in $\mathbb{R}^{100}$. The "easy" distribution $\mathcal{D}_1$ has margin $\gamma = 3$ and low variance $\sigma = 0.5$, while the hard distribution $\mathcal{D}_2$ varies over $\gamma \in \{0.1, 0.5, 1.0, 2.0\}$ and $\sigma \in \{0.5, 1.0, 1.5, 2.0\}$. We generate 1K training samples from $\mathcal{D}_1, \mathcal{D}_2$.

Linear classifiers are trained using hinge loss and gradient descent (2K epochs, learning rate from $0.001, \ldots, 1.0$). The baseline trains only on $\mathcal{D}_2$, while our curriculum method (Algorithm 2) first trains on $\mathcal{D}_1$ and then fine-tunes on $\mathcal{D}_2$ with $\ell_2$ regularization $\lambda\|\mathrm{w}_2 - \widehat{\mathrm{w}}_1\|^2$, where $\widehat{\mathrm{w}}_1$ is the solution from the first stage. $\lambda$ is selected from $\{10^{-5}, 10^{-4}, 10^{-3}, 10^{-2}, 10^{-1}, 1, 10\}$ using validation data.

Each experiment is repeated 10 times, and we report mean test accuracy and standard deviation in Figure 2. Curriculum learning consistently outperforms the baseline, demonstrating that starting with an easier task aids learning on harder ones. The performance gap widens as the target task becomes more difficult–i.e., with smaller margins and higher variance–highlighting the effectiveness of the curriculum approach under challenging conditions.

Next, we apply our theory and methods to adversarially robust learning. In adversarial robustness, an adversary perturbs an input x within a perturbation set $\mathcal{B}(\mathrm{x})$, and the standard loss $\ell_t((\mathrm{x}, y); \mathrm{w})$ is replaced by the robust loss: $\ell_t^{rob}((\mathrm{x}, y); \mathrm{w}) := \sup_{\widetilde{\mathrm{x}} \in \mathcal{B}(\mathrm{x})} \ell_t((\widetilde{\mathrm{x}}, y); \mathrm{w})$. This replacement preserves convexity and Lipschitz continuity (see Appendix B.5), allowing us to extend the results of Sections 4.1–4.3 to the robustness setting. In Algorithm 3, the subgradient $\nabla_\mathrm{w}\ell_t^{\mathrm{rob}}(\mathrm{w}_{k-1}; z_k)$ is computed using adversarial training techniques.

However, smoothness does not generally carry over: while the standard loss may be smooth, the robust loss is known to be non-smooth [Xing et al., 2021]. Thus, Theorem 4.6 cannot be directly

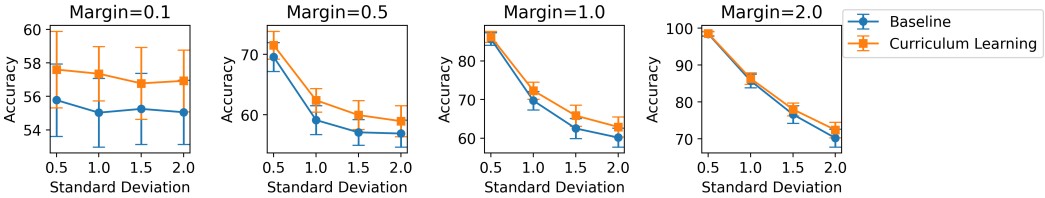

Figure 2: Test accuracy as a function of standard deviation for different margin $\gamma$.

applied. Nevertheless, we show (Appendix B.5) that if the standard loss is nonnegative and $H_t$-smooth, then Theorem 4.6 still holds for the robust loss. This insight allows curriculum learning results to carry over to adversarial settings simply by substituting standard loss with robust loss. In practice, good bias/initialization for robust training can come from a non-robust model, a model trained with weaker attacks, or a related task.

We evaluate curriculum adversarial training with $\ell_2$ regularization on MNIST dataset. Adversarial examples are generated using 10-step PGD with step size $\alpha/5$ under an $\ell_\infty$ perturbation budget $\alpha \in \{0.1, 0.2, 0.3, 0.4\}$. For curriculum training, we define task $t$ with attack strength $\alpha t/T$, for $t \in [T]$ and $T \in \{1, 2, 3\}$. No regularization is used for $t = 1$. From $t \geq 2$, we incorporate $\ell_2$ regularization of the form $\lambda \|w_t - \widehat{w}_{t-1}\|^2$, where $\widehat{w}_{t-1}$ is the previous model and $\lambda \in \{10^{-5}, 10^{-4}, 10^{-3}, 10^{-2}\}$.

We use a CNN with two convolutional layers followed by max-pooling and two fully connected layers with ReLU activations. The conv layers use [input, output, kernel] = [1, 10, 5] and [10, 20, 5]; the fully connected layers have dimensions [320, 100] and [100, 10]. Models are trained with cross-entropy loss using Adam for 100 epochs, batch size 128, and learning rate chosen from $\{10^{-4}, 10^{-3}, 10^{-2}, 10^{-1}\}$. Early stopping is used based on robust validation accuracy (measured with PGD attack of size $\alpha$) to select both the model and hyperparameters.

We report both standard and robust test accuracy under PGD attack of size $\alpha$ in Table 1, averaged over three runs with standard deviation. We note that curriculum adversarial training maintains performance for small $\alpha$ and provides notable improvements for larger $\alpha$ values–particularly when $\alpha \geq 0.3$. This supports the hypothesis that initializing from easier tasks (weaker attacks) enhances robustness against stronger adversaries.

For additional experimental details and extended results, please see the supplementary material.

| $\alpha$ \ T | 1 | | 2 | | 3 | |
|---|---|---|---|---|---|---|
| | nat acc | pgd acc | nat acc | pgd acc | nat acc | pgd acc |
| 0.1 | 99.18±0.07 | 96.07±0.02 | 99.27±0.07 | 95.65±0.18 | 99.36±0.03 | 95.74±0.14 |
| 0.2 | 98.80±0.03 | 94.73±0.22 | 98.86±0.15 | 94.60±0.93 | 98.67±0.05 | 94.38±0.23 |
| 0.3 | 98.27±0.46 | 92.77±1.20 | 98.77±0.15 | 94.74±0.12 | 98.23±0.15 | 93.61±0.87 |
| 0.4 | 11.35±0.00 | 11.35±0.00 | 98.39±0.29 | 95.54±0.41 | 98.52±0.14 | 95.63±0.12 |

Table 1: Standard (nat acc) / robust (pgd acc) accuracy under $\ell_\infty$ PGD attack of size $\alpha$ (MNIST).

## 7 Conclusion

In this work, we provide theoretical guarantees for both convex and nonconvex learning problems under a multi-task curriculum learning framework that leverages implicit bias from prior tasks. Central to our analysis is the proposed $(r, \alpha)$ condition, which characterizes a 'good' curriculum by quantifying task similarity and enabling reduced sample complexity. While the $(r, \alpha)$ condition offers a principled way to evaluate curriculum quality, it may be difficult to verify in practice. A promising direction for future work is to investigate when this condition holds for specific problem families and how it can guide the design of effective, data-driven curricula.

## Acknowledgments

This research was supported, in part, by the NSF CAREER award IIS-1943251.

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

# A  Missing Proofs in Section 3

**Lemma A.1.** $\mathbb{E}_{S_1 \sim \mathcal{D}_1^{n_1}} \left[ \|\widehat{w}_1 - w_1^\star\|_2^2 \right] \leq \frac{\rho_1^2}{\lambda^2 n_1}$.

*Proof of Lemma A.1.* Denote $g_1(z; w) = \nabla_w \ell_1(z; w)$. The gradient of the population loss can be written as

$$0 = \nabla_w L_{\mathcal{D}_1}(w_1^\star) = \nabla_w \mathbb{E}_{z \sim \mathcal{D}_1} \ell_1(z; w_1^\star) = \mathbb{E}_{z \sim \mathcal{D}_1} \nabla_w \ell_1(z; w_1^\star) = \mathbb{E}_{z \sim \mathcal{D}_1} g_1(z; w_1^\star).$$

This leads to

$$\mathbb{E}_{S_1 \sim \mathcal{D}_1^{n_1}} \|\nabla_w \widehat{L}_{S_1}(w_1^\star)\|_2^2$$
$$= \mathbb{E}_{S_1 \sim \mathcal{D}_1^{n_1}} \|\nabla_w \widehat{L}_{S_1}(w_1^\star) - \nabla_w L_{\mathcal{D}_1}(w_1^\star)\|_2^2$$
$$= \mathbb{E}_{S_1 \sim \mathcal{D}_1^{n_1}} \left\| \frac{1}{n_1} \sum_{\widehat{z} \in S_1} g_1(\widehat{z}; w_1^\star) - \mathbb{E}_{z \sim \mathcal{D}_1} g_1(z; w_1^\star) \right\|_2^2$$
$$= \mathbb{E}_{S_1 \sim \mathcal{D}_1^{n_1}} \frac{1}{n_1^2} \sum_{\widehat{z} \in S_1} \|g_1(\widehat{z}; w_1^\star) - \mathbb{E}_{z \sim \mathcal{D}_1} g_1(z; w_1^\star)\|_2^2$$
$$= \frac{1}{n_1} \mathbb{E}_{\widehat{z} \sim \mathcal{D}_1} \|g_1(\widehat{z}; w_1^\star) - \mathbb{E}_{z \sim \mathcal{D}_1} g_1(z; w_1^\star)\|_2^2$$
$$= \frac{1}{n_1} \left( \mathbb{E}_{\widehat{z} \sim \mathcal{D}_1} \|g_1(\widehat{z}; w_1^\star)\|_2^2 - \|\mathbb{E}_{z \sim \mathcal{D}_1} g_1(z; w_1^\star)\|_2^2 \right) \leq \frac{\rho_1^2}{n_1}.$$

Since $\widehat{L}_{S_1}(w)$ is $\lambda$-strongly convex,

$$\|\nabla_w \widehat{L}_{S_1}(w_1^\star)\|_2 = \|\nabla_w \widehat{L}_{S_1}(w_1^\star) - \nabla_w \widehat{L}_{S_1}(\widehat{w}_1)\|_2 \geq \lambda \|w_1^\star - \widehat{w}_1\|_2.$$

Therefore,

$$\mathbb{E}_{S_1 \sim \mathcal{D}_1^{n_1}} \left[ \|\widehat{w}_1 - w_1^\star\|_2^2 \right] \leq \frac{1}{\lambda^2} \mathbb{E}_{S_1 \sim \mathcal{D}_1^{n_1}} \|\nabla_w \widehat{L}_{S_1}(w_1^\star)\|_2^2 \leq \frac{\rho_1^2}{\lambda^2 n_1}.$$

$\square$

**Theorem 3.1.** If the second task is convex, then setting $\mu_2 = \frac{2\rho_2}{\left( \|w_2^\star - w_1^\star\|_2 + \frac{\rho_1}{\lambda \sqrt{n_1}} \right) \sqrt{n_2}}$, we have

$\mathbb{E}\left[ \varepsilon_2(\widehat{w}_2) \right] \leq \frac{2\rho_2}{\sqrt{n_2}} \left( \|w_2^\star - w_1^\star\|_2 + \frac{\rho_1}{\lambda \sqrt{n_1}} \right)$.

*Proof of Theorem 3.1.* From the theory of RERM in Shalev-Shwartz and Ben-David [2014] Chapter 13, if $S_1$ is fixed, the second phase RERM is $\frac{2\rho_2^2}{\mu_2 n_2}$-uniformly stable if only one data in $S_2$ is replaced. Therefore,

$$\mathbb{E}_{S_2} L_{\mathcal{D}_2}(\widehat{w}_2) \leq \mathbb{E}_{S_2} \widehat{L}_{S_2}(\widehat{w}_2) + \frac{2\rho_2^2}{\mu_2 n_2}$$
$$\leq \mathbb{E}_{S_2} \left[ \widehat{L}_{S_2}(\widehat{w}_2) + \frac{\mu_2}{2} \|\widehat{w}_2 - \widehat{w}_1\|_2^2 \right] + \frac{2\rho_2^2}{\mu_2 n_2}$$
$$\leq \mathbb{E}_{S_2} \left[ \widehat{L}_{S_2}(w_2^\star) + \frac{\mu_2}{2} \|w_2^\star - \widehat{w}_1\|_2^2 \right] + \frac{2\rho_2^2}{\mu_2 n_2} \qquad \text{(from the definition of RERM)}$$
$$= L_{\mathcal{D}_2}(w_2^\star) + \frac{\mu_2}{2} \|w_2^\star - \widehat{w}_1\|_2^2 + \frac{2\rho_2^2}{\mu_2 n_2}.$$

Taking expectation w.r.t. $S_1 \sim \mathcal{D}_1^{n_1}$,

$$\mathbb{E}_{S_1,S_2} L_{\mathcal{D}_2}(\widehat{w}_2) \leq L_{\mathcal{D}_2}(w_2^\star) + \frac{\mu_2}{2} \mathbb{E}_{S_1} \|w_2^\star - \widehat{w}_1\|_2^2 + \frac{2\rho_2^2}{\mu_2 n_2}$$

$$\leq L_{\mathcal{D}_2}(w_2^\star) + \frac{\mu_2}{2} \mathbb{E}_{S_1} (\|w_2^\star - w_1^\star\|_2 + \|\widehat{w}_1 - w_1^\star\|_2)^2 + \frac{2\rho_2^2}{\mu_2 n_2}$$
$$\text{(triangle inequality)}$$

$$\leq L_{\mathcal{D}_2}(w_2^\star) + \frac{\mu_2}{2} \left( \|w_2^\star - w_1^\star\|_2 + \frac{\rho_1}{\lambda\sqrt{n_1}} \right)^2 + \frac{2\rho_2^2}{\mu_2 n_2}. \qquad \text{(Lemma A.1)}$$

Setting $\mu_2 = \frac{2\rho_2}{\left( \|w_2^\star - w_1^\star\|_2 + \frac{\rho_1}{\lambda\sqrt{n_1}} \right)\sqrt{n_2}}$, we obtain

$$\mathbb{E}\left[ L_{\mathcal{D}_2}(\widehat{w}_2) \right] \leq L_{\mathcal{D}_2}(w^\star) + \frac{2\rho_2}{\sqrt{n_2}} \left( \|w_2^\star - w_1^\star\|_2 + \frac{\rho_1}{\lambda\sqrt{n_1}} \right).$$

$\square$

**Theorem 3.2.** If the second task is $l$-weakly convex, then setting $\mu_2 = l + \frac{2\rho_2}{\left( \|w_2^\star - w_1^\star\|_2 + \frac{\rho_1}{\lambda\sqrt{n_1}} \right)\sqrt{n_2}}$, we have $\mathbb{E}\left[ \varepsilon_2(\widehat{w}_2) \right] \leq \frac{2\rho_2}{\sqrt{n_2}} \left( \|w_2^\star - w_1^\star\|_2 + \frac{\rho_1}{\lambda\sqrt{n_1}} \right) + \frac{l}{2} \left( \|w_2^\star - w_1^\star\|_2 + \frac{\rho_1}{\lambda\sqrt{n_1}} \right)^2$.

*Proof of Theorem 3.2.* If $S_1$ is fixed, for any $\mu_2 > l$, the regularized loss $\ell_2(z; w) + \frac{\mu_2}{2}\|w - \widehat{w}_1\|_2^2$ is $(\mu_2 - l)$-strongly convex. From the theory of RERM in Shalev-Shwartz and Ben-David [2014] Chapter 13, the second phase RERM is $\frac{2\rho_2^2}{(\mu_2 - l)n_2}$-uniformly stable if only one data in $S_2$ is replaced. Therefore,

$$\mathbb{E}_{S_2} L_{\mathcal{D}_2}(\widehat{w}_2) \leq \mathbb{E}_{S_2} \widehat{L}_{S_2}(\widehat{w}_2) + \frac{2\rho_2^2}{(\mu_2 - l)n_2}$$

$$\leq \mathbb{E}_{S_2} \left[ \widehat{L}_{S_2}(\widehat{w}_2) + \frac{\mu_2}{2} \|\widehat{w}_2 - \widehat{w}_1\|_2^2 \right] + \frac{2\rho_2^2}{(\mu_2 - l)n_2}$$

$$\leq \mathbb{E}_{S_2} \left[ \widehat{L}_{S_2}(w_2^\star) + \frac{\mu_2}{2} \|w_2^\star - \widehat{w}_1\|_2^2 \right] + \frac{2\rho_2^2}{(\mu_2 - l)n_2} \quad \text{(from the definition of RERM)}$$

$$= L_{\mathcal{D}_2}(w_2^\star) + \frac{\mu_2}{2} \|w_2^\star - \widehat{w}_1\|_2^2 + \frac{2\rho_2^2}{(\mu_2 - l)n_2}.$$

Taking expectation w.r.t. $S_1 \sim \mathcal{D}_1^{n_1}$,

$$\mathbb{E}_{S_1,S_2} L_{\mathcal{D}_2}(\widehat{w}_2) \leq L_{\mathcal{D}_2}(w_2^\star) + \frac{\mu_2}{2} \mathbb{E}_{S_1} \|w_2^\star - \widehat{w}_1\|_2^2 + \frac{2\rho_2^2}{(\mu_2 - l)n_2}$$

$$\leq L_{\mathcal{D}_2}(w_2^\star) + \frac{\mu_2}{2} \mathbb{E}_{S_1} (\|w_2^\star - w_1^\star\|_2 + \|\widehat{w}_1 - w_1^\star\|_2)^2 + \frac{2\rho_2^2}{(\mu_2 - l)n_2}$$
$$\text{(triangle inequality)}$$

$$\leq L_{\mathcal{D}_2}(w_2^\star) + \frac{\mu_2}{2} \left( \|w_2^\star - w_1^\star\|_2 + \frac{\rho_1}{\lambda\sqrt{n_1}} \right)^2 + \frac{2\rho_2^2}{(\mu_2 - l)n_2}. \qquad \text{(Lemma A.1)}$$

Setting $\mu_2 = l + \frac{2\rho_2}{\left( \|w_2^\star - w_1^\star\|_2 + \frac{\rho_1}{\lambda\sqrt{n_1}} \right)\sqrt{n_2}}$, we obtain

$$\mathbb{E}\left[ L_{\mathcal{D}_2}(\widehat{w}_2) \right] \leq L_{\mathcal{D}_2}(w^\star) + \frac{2\rho_2}{\sqrt{n_2}} \left( \|w_2^\star - w_1^\star\|_2 + \frac{\rho_1}{\lambda\sqrt{n_1}} \right) + \frac{l}{2} \left( \|w_2^\star - w_1^\star\|_2 + \frac{\rho_1}{\lambda\sqrt{n_1}} \right)^2.$$

$\square$

# B    Missing Details in Section 4

## B.1    Missing Proofs in Section 4.1

**Theorem 4.1.** Suppose task $t$ is convex and $\rho_t$-Lipschitz, and the $(r_t, \alpha)$ condition holds between tasks $(t-1, t)$. Then, setting $\mu_t = \frac{2\rho_t}{r_t\sqrt{n_t}}$ yields the following excess risk bound:

$$\mathbb{E}\left[\varepsilon_t(\widehat{w}_t)\right] \leq \frac{2r_t\rho_t}{\sqrt{n_t}} + \alpha\mathbb{E}\left[\varepsilon_{t-1}(\widehat{w}_{t-1})\right].$$

*Proof of Theorem 4.1.* If $S_1, \ldots, S_{t-1}$ is fixed, for any $\mu_t > 0$, the regularized loss $\ell_t(z; w) + \frac{\mu_t}{2}\|w - \widehat{w}_{t-1}\|_2^2$ is $\mu_t$-strongly convex. From the theory of RERM in Shalev-Shwartz and Ben-David [2014] Chapter 13, the $t$-th step RERM is $\frac{2\rho_t^2}{\mu_t n_t}$-uniformly stable if only one data in $S_t$ is replaced. Therefore, $\forall w' \in \mathbb{R}^m$ independent with $S_1, \ldots, S_t$,

$$\begin{aligned}
\mathbb{E}_{S_t} L_{\mathcal{D}_t}(\widehat{w}_t) &\leq \mathbb{E}_{S_t}\widehat{L}_{S_t}(\widehat{w}_t) + \frac{2\rho_t^2}{\mu_t n_t} \\
&\leq \mathbb{E}_{S_t}\left[\widehat{L}_{S_t}(\widehat{w}_t) + \frac{\mu_t}{2}\|\widehat{w}_t - \widehat{w}_{t-1}\|_2^2\right] + \frac{2\rho_t^2}{\mu_t n_t} \\
&\leq \mathbb{E}_{S_t}\left[\widehat{L}_{S_t}(w') + \frac{\mu_t}{2}\|w' - \widehat{w}_{t-1}\|_2^2\right] + \frac{2\rho_t^2}{\mu_t n_t} \qquad \text{(from the definition of RERM)} \\
&= L_{\mathcal{D}_t}(w') + \frac{\mu_t}{2}\|w' - \widehat{w}_{t-1}\|_2^2 + \frac{2\rho_t^2}{\mu_t n_t}.
\end{aligned}$$

Since task $t-1$ and task $t$ satisfy $(r_t, \alpha)$ condition, there exists $w'$, s.t. $\|w' - \widehat{w}_{t-1}\|_2 \leq r_t$ and $\varepsilon_t(w') \leq \alpha\varepsilon_{t-1}(\widehat{w}_{t-1})$ hold. Thus,

$$\begin{aligned}
\mathbb{E}_{S_t} L_{\mathcal{D}_t}(\widehat{w}_t) &\leq L_{\mathcal{D}_t}(w') + \frac{\mu_t}{2}\|w' - \widehat{w}_{t-1}\|_2^2 + \frac{2\rho_t^2}{\mu_t n_t} \\
&= \inf_w L_{\mathcal{D}_t}(w) + \varepsilon_t(w') + \frac{\mu_t}{2}\|w' - \widehat{w}_{t-1}\|_2^2 + \frac{2\rho_t^2}{\mu_t n_t} \\
&\leq \inf_w L_{\mathcal{D}_t}(w) + \alpha\varepsilon_{t-1}(\widehat{w}_{t-1}) + \frac{\mu_t r_t^2}{2} + \frac{2\rho_t^2}{\mu_t n_t}.
\end{aligned}$$

Setting $\mu_t = \frac{2\rho_t}{r_t\sqrt{n_t}}$, we have $\mathbb{E}_{S_t}\varepsilon_t(\widehat{w}_t) \leq \alpha\varepsilon_{t-1}(\widehat{w}_{t-1}) + \frac{2r_t\rho_t}{\sqrt{n_t}}$. Taking expectation w.r.t. $S_1, \ldots, S_{t-1}$, we obtain

$$\mathbb{E}\left[\varepsilon_t(\widehat{w}_t)\right] \leq \frac{2r_t\rho_t}{\sqrt{n_t}} + \alpha\mathbb{E}\left[\varepsilon_{t-1}(\widehat{w}_{t-1})\right].$$

$\square$

**Corollary 4.2.** Assume the first task is learned with excess risk $\mathbb{E}\left[\varepsilon_1(\widehat{w}_1)\right] \leq \epsilon$. Set the regularization parameter to $\mu_t = \frac{2\rho_t}{r_t\sqrt{n_t}}$, and suppose the sample size $n_t \geq \frac{4r_t^2\rho_t^2}{(1-\alpha)^2\epsilon^2}$. Then, for all tasks $t$, the excess risk is bounded as $\mathbb{E}\left[\varepsilon_t(\widehat{w}_t)\right] \leq \epsilon$.

*Proof of Corollary 4.2.* Theorem 4.1 gives

$$\mathbb{E}\left[\varepsilon_t(\widehat{w}_t)\right] \leq \frac{2r_t\rho_t}{\sqrt{n_t}} + \alpha\mathbb{E}\left[\varepsilon_{t-1}(\widehat{w}_{t-1})\right] \leq \alpha\mathbb{E}\left[\varepsilon_{t-1}(\widehat{w}_{t-1})\right] + (1-\alpha)\epsilon.$$

We can use induction to prove that $\mathbb{E}\left[\varepsilon_t(\widehat{w}_t)\right] \leq \epsilon$. $\square$

**Corollary 4.3.** Suppose the first task is learned to excess risk $\mathbb{E}\left[\varepsilon_1(\widehat{w}_1)\right] \leq \epsilon_1$. Set the regularization parameter as $\mu_t = \frac{2\rho_t}{r_t\sqrt{n_t}}$, and assume the sample size satisfies $n_t \geq \left(\frac{4r_t\rho_t}{(1-\alpha)\epsilon_1\left(\frac{\alpha+1}{2}\right)^{t-2}}\right)^2$. Then, for every task $t$, we have $\mathbb{E}\left[\varepsilon_t(\widehat{w}_t)\right] \leq \epsilon_1\left(\frac{\alpha+1}{2}\right)^{t-1}$.

*Proof of Corollary 4.3.* Theorem 4.1 gives

$$\mathbb{E}\left[\varepsilon_t(\widehat{w}_t)\right] \leq \frac{2r_t\rho_t}{\sqrt{n_t}} + \alpha\mathbb{E}\left[\varepsilon_{t-1}(\widehat{w}_{t-1})\right] \leq \alpha\mathbb{E}\left[\varepsilon_{t-1}(\widehat{w}_{t-1})\right] + \frac{1-\alpha}{2}\epsilon_1\left(\frac{\alpha+1}{2}\right)^{t-2}.$$

We can use induction to prove that $\mathbb{E}\left[\varepsilon_t(\widehat{w}_t)\right] \leq \epsilon_1\left(\frac{\alpha+1}{2}\right)^{t-1}$. $\qquad\square$

## B.2 Missing Proofs in Section 4.2

**Theorem 4.4.** Suppose task $t$ has a $\rho_t$-Lipschitz convex loss function and satisfies the $(r_t, \alpha)$ condition with task $t-1$. Choosing the learning rate $\eta_t = \frac{r_t}{\rho_t\sqrt{n_t}}$, the excess risk of SGD satisfies

$$\mathbb{E}\left[\varepsilon_t(\widehat{w}_t)\right] \leq \frac{r_t\rho_t}{\sqrt{n_t}} + \alpha\mathbb{E}\left[\varepsilon_{t-1}(\widehat{w}_{t-1})\right].$$

*Proof of Theorem 4.4.* Let's first fix $S_1, \ldots, S_{t-1}$. Since task $t-1$ and task $t$ satisfy $(r_t, \alpha)$ condition, there exists $w'$, s.t. $\|w' - \widehat{w}_{t-1}\|_2 \leq r_t$ and $\varepsilon_t(w') \leq \alpha\varepsilon_{t-1}(\widehat{w}_{t-1})$ hold. For $k = 1, 2, \ldots, n_t$,

$$\begin{aligned}
\|w_k - w'\|_2^2 &= \|w_{k-1} - w' - \eta_t\nabla_w\ell_t(w_{k-1}; z_k)\|_2^2 \\
&= \|w_{k-1} - w'\|_2^2 + \eta_t^2\|\nabla_w\ell_t(w_{k-1}; z_k)\|_2^2 + 2\eta_t\langle w' - w_{k-1}, \nabla_w\ell_t(w_{k-1}; z_k)\rangle \\
&\leq \|w_{k-1} - w'\|_2^2 + \eta_t^2\rho_t^2 + 2\eta_t\left(\ell_t(w'; z_k) - \ell_t(w_{k-1}; z_k)\right).
\end{aligned}$$
$$\text{(Lipschitz and convex loss)}$$

Rewriting this inequality gives

$$\ell_t(w_{k-1}; z_k) \leq \ell_t(w'; z_k) + \frac{\eta_t\rho_t^2}{2} + \frac{\|w_{k-1} - w'\|_2^2 - \|w_k - w'\|_2^2}{2\eta_t}$$

Taking average over $k$, we get

$$\begin{aligned}
\frac{1}{n_t}\sum_{k=1}^{n_t}\ell_t(w_{k-1}; z_k) &\leq \frac{1}{n_t}\sum_{k=1}^{n_t}\ell_t(w'; z_k) + \frac{\eta_t\rho_t^2}{2} + \frac{\|\widehat{w}_{t-1} - w'\|_2^2 - \|w_{n_t} - w'\|_2^2}{2\eta_t n_t} \\
&\leq \frac{1}{n_t}\sum_{k=1}^{n_t}\ell_t(w'; z_k) + \frac{\eta_t\rho_t^2}{2} + \frac{r_t^2}{2\eta_t n_t} \\
&= \frac{1}{n_t}\sum_{k=1}^{n_t}\ell_t(w'; z_k) + \frac{r_t\rho_t}{\sqrt{n_t}}
\end{aligned}$$

Since $z_k$ is independent with $w_{k-1}$, taking expectation w.r.t. $S_t \sim \mathcal{D}_t^{n_t}$ gives

$$\begin{aligned}
\frac{1}{n_t}\sum_{k=1}^{n_t}\mathbb{E}_{S_t}\left[L_{\mathcal{D}_t}(w_{k-1})\right] &= \frac{1}{n_t}\sum_{k=1}^{n_t}\mathbb{E}_{S_t}\left[\ell_t(w_{k-1}; z_k)\right] \\
&\leq \frac{1}{n_t}\sum_{k=1}^{n_t}\mathbb{E}_{S_t}\left[\ell_t(w'; z_k)\right] + \frac{r_t\rho_t}{\sqrt{n_t}} \\
&= L_{\mathcal{D}_t}(w') + \frac{r_t\rho_t}{\sqrt{n_t}}.
\end{aligned}$$

Using Jensen's Inequality,

$$\begin{aligned}
\mathbb{E}_{S_t}\left[\varepsilon_t(\widehat{w}_t)\right] &= \mathbb{E}_{S_t}\left[L_{\mathcal{D}_t}(\widehat{w}_t)\right] - \inf_w L_{\mathcal{D}_t}(w) \\
&\leq \frac{1}{n_t}\sum_{k=1}^{n_t}\mathbb{E}_{S_t}\left[L_{\mathcal{D}_t}(w_{k-1})\right] - \inf_w L_{\mathcal{D}_t}(w) \\
&\leq L_{\mathcal{D}_t}(w') + \frac{r_t\rho_t}{\sqrt{n_t}} - \inf_w L_{\mathcal{D}_t}(w) \\
&\leq \frac{r_t\rho_t}{\sqrt{n_t}} + \alpha\varepsilon_{t-1}(\widehat{w}_{t-1}).
\end{aligned}$$

Taking expectation w.r.t. $S_1, \ldots, S_{t-1}$, we obtain

$$\mathbb{E}\left[\varepsilon_t(\widehat{w}_t)\right] \leq \frac{r_t\rho_t}{\sqrt{n_t}} + \alpha\mathbb{E}\left[\varepsilon_{t-1}(\widehat{w}_{t-1})\right].$$

$\qquad\square$

## B.3 Missing Proofs in Section 4.3

**Theorem 4.5.** Choosing $\mu_t$ appropriately, the excess risk of curriculum learning satisfies

$$\mathbb{E}\left[\varepsilon_t(\widehat{\mathbf{w}}_t)\right] \leq \frac{2r_t}{\sqrt{n_t}}\left(\bar{\rho}_t + \frac{6r_t(\rho_t^2 - \bar{\rho}_t^2)}{(1-\alpha)\bar{\varepsilon}_t\sqrt{n_t}}\right) + \frac{1+\alpha}{2}\mathbb{E}\left[\varepsilon_{t-1}(\widehat{\mathbf{w}}_{t-1})\right].$$

*Proof of Theorem 4.5.* Let $\mu_t$ be a constant to be determined.
$p_0 := \mathbb{P}_{S_t \sim \mathcal{D}_t^{n_t}}\left(L_{\mathcal{D}_t}(\widehat{\mathbf{w}}_t) - \inf_{\mathbf{w}} L_{\mathcal{D}_t}(\mathbf{w}) > \bar{\varepsilon}_t\right)$. Recall $\mathbb{E}_{S_t}\left[\varepsilon_t(\widehat{\mathbf{w}}_t)\right] = \mathbb{E}_{S_t \sim \mathcal{D}_t^{n_t}} L_{\mathcal{D}_t}(\widehat{\mathbf{w}}_t) - \inf_{\mathbf{w}} L_{\mathcal{D}_t}(\mathbf{w})$. Using Markov's Inequality,

$$p_0 \leq \frac{\mathbb{E}_{S_t}\left[\varepsilon_t(\widehat{\mathbf{w}}_t)\right]}{\bar{\varepsilon}_t}. \tag{5}$$

Let $S_t = \{z_1, z_2, \ldots, z_{n_t}\} \sim \mathcal{D}_t^{n_t}$ and $S_t' = \{z_1', z_2, \ldots, z_{n_t}\} \sim \mathcal{D}_t^{n_t}$ be two neighboring data sets that differ in one single example. $S_t \cup S_t' = \{z_1', z_1, z_2, \ldots, z_{n_t}\} \sim \mathcal{D}^{n_t+1}$.
Recall $\widehat{\mathbf{w}}_t \in \operatorname{argmin}_{\mathbf{w}}\left(\widehat{L}_{S_t}(\mathbf{w}) + \frac{\mu_t}{2}\|\mathbf{w} - \widehat{\mathbf{w}}_{t-1}\|_2^2\right)$; $\widehat{\mathbf{w}}_t' \in \operatorname{argmin}_{\mathbf{w}}\left(\widehat{L}_{S_t'}(\mathbf{w}) + \frac{\mu_t}{2}\|\mathbf{w} - \widehat{\mathbf{w}}_{t-1}\|_2^2\right)$.

Since the optimization objective $\widehat{L}_{S_t}(\mathbf{w}) + \frac{\mu_t}{2}\|\mathbf{w} - \widehat{\mathbf{w}}_{t-1}\|_2^2$ is $\mu_t$-strongly convex, we have

$$\widehat{L}_{S_t}(\widehat{\mathbf{w}}_t') + \frac{\mu_t}{2}\|\widehat{\mathbf{w}}_t' - \widehat{\mathbf{w}}_{t-1}\|_2^2 \geq \widehat{L}_{S_t}(\widehat{\mathbf{w}}_t) + \frac{\mu_t}{2}\|\widehat{\mathbf{w}}_t - \widehat{\mathbf{w}}_{t-1}\|_2^2 + \frac{\mu_t}{2}\|\widehat{\mathbf{w}}_t' - \widehat{\mathbf{w}}_t\|_2^2. \tag{6}$$

Similarly,

$$\widehat{L}_{S_t'}(\widehat{\mathbf{w}}_t) + \frac{\mu_t}{2}\|\widehat{\mathbf{w}}_t - \widehat{\mathbf{w}}_{t-1}\|_2^2 \geq \widehat{L}_{S_t'}(\widehat{\mathbf{w}}_t') + \frac{\mu_t}{2}\|\widehat{\mathbf{w}}_t' - \widehat{\mathbf{w}}_{t-1}\|_2^2 + \frac{\mu_t}{2}\|\widehat{\mathbf{w}}_t' - \widehat{\mathbf{w}}_t\|_2^2. \tag{7}$$

Adding up equation (6) and equation (7),

$$\mu_t\|\widehat{\mathbf{w}}_t' - \widehat{\mathbf{w}}_t\|_2^2 \leq \frac{\ell_t(z_1; \widehat{\mathbf{w}}_t') - \ell_t(z_1; \widehat{\mathbf{w}}_t)}{n_t} + \frac{\ell_t(z_1'; \widehat{\mathbf{w}}_t) - \ell_t(z_1'; \widehat{\mathbf{w}}_t')}{n_t}. \tag{8}$$

We say $S_t \cup S_t'$ is good if $L_{\mathcal{D}_t}(\widehat{\mathbf{w}}_t) - \inf_{\mathbf{w}} L_{\mathcal{D}_t}(\mathbf{w}) \leq \bar{\varepsilon}_t$ and $L_{\mathcal{D}_t}(\widehat{\mathbf{w}}_t') - \inf_{\mathbf{w}} L_{\mathcal{D}_t}(\mathbf{w}) \leq \bar{\varepsilon}_t$ hold simultaneously. Otherwise, we say $S_t \cup S_t'$ is bad. Applying a union bound and combining with equation (5),

$$\mathbb{P}_{S_t \cup S_t' \sim \mathcal{D}_t^{n_t+1}}\left(S_t \cup S_t' \text{ is bad}\right) \leq 2p_0 \leq \frac{2\mathbb{E}_{S_t}\left[\varepsilon_t(\widehat{\mathbf{w}}_t)\right]}{\bar{\varepsilon}_t}. \tag{9}$$

If $S_t \cup S_t'$ is good, by the assumption on the local Lipschitz constant, $|\ell_t(z; \widehat{\mathbf{w}}_t) - \ell_t(z; \widehat{\mathbf{w}}_t')| \leq \bar{\rho}_t\|\widehat{\mathbf{w}}_t' - \widehat{\mathbf{w}}_t\|_2$ holds for any z. Equation (8) implies

$$\mu_t\|\widehat{\mathbf{w}}_t' - \widehat{\mathbf{w}}_t\|_2^2 \leq \frac{\ell_t(z_1; \widehat{\mathbf{w}}_t') - \ell_t(z_1; \widehat{\mathbf{w}}_t)}{n_t} + \frac{\ell_t(z_1'; \widehat{\mathbf{w}}_t) - \ell_t(z_1'; \widehat{\mathbf{w}}_t')}{n_t} \leq \frac{2\bar{\rho}_t}{n_t}\|\widehat{\mathbf{w}}_t' - \widehat{\mathbf{w}}_t\|_2.$$

Therefore, $\|\widehat{\mathbf{w}}_t' - \widehat{\mathbf{w}}_t\|_2 \leq \frac{2\bar{\rho}_t}{\mu_t n_t}$ if $S_t \cup S_t'$ is good. Thus, we also know that $|\ell_t(z; \widehat{\mathbf{w}}_t) - \ell_t(z; \widehat{\mathbf{w}}_t')| \leq \bar{\rho}_t\|\widehat{\mathbf{w}}_t' - \widehat{\mathbf{w}}_t\|_2 \leq \frac{2\bar{\rho}_t^2}{\mu_t n_t}$ holds for any z. If $S_t \cup S_t'$ is bad, using the global Lipschitz constant, $|\ell_t(z; \widehat{\mathbf{w}}_t) - \ell_t(z; \widehat{\mathbf{w}}_t')| \leq \rho_t\|\widehat{\mathbf{w}}_t' - \widehat{\mathbf{w}}_t\|_2$ holds for any z. We similarly get $\|\widehat{\mathbf{w}}_t' - \widehat{\mathbf{w}}_t\|_2 \leq \frac{2\rho_t}{\mu_t n_t}$ if $S_t \cup S_t'$ is bad. We also know that $|\ell_t(z; \widehat{\mathbf{w}}_t) - \ell_t(z; \widehat{\mathbf{w}}_t')| \leq \rho_t\|\widehat{\mathbf{w}}_t' - \widehat{\mathbf{w}}_t\|_2 \leq \frac{2\rho_t^2}{\mu_t n_t}$ is true for any z. Now we upper bound the generalization gap of RERM:

$$\begin{aligned}
&\mathbb{E}_{S_t \sim \mathcal{D}_t^{n_t}}\left(L_{\mathcal{D}_t}(\widehat{\mathbf{w}}_t) - \widehat{L}_{S_t}(\widehat{\mathbf{w}}_t)\right) \\
=&\mathbb{E}_{S_t \cup S_t' \sim \mathcal{D}_t^{n_t+1}}\left(\ell_t(z_1; \widehat{\mathbf{w}}_t') - \ell_t(z_1; \widehat{\mathbf{w}}_t)\right) \\
\leq&\frac{2\bar{\rho}_t^2}{\mu_t n_t}\mathbb{P}_{S_t \cup S_t' \sim \mathcal{D}_t^{n_t+1}}\left(S_t \cup S_t' \text{ is good}\right) + \frac{2\rho_t^2}{\mu_t n_t}\mathbb{P}_{S_t \cup S_t' \sim \mathcal{D}_t^{n_t+1}}\left(S_t \cup S_t' \text{ is bad}\right) \\
=&\frac{2\bar{\rho}_t^2}{\mu_t n_t} + \frac{2(\rho_t^2 - \bar{\rho}_t^2)}{\mu_t n_t}\mathbb{P}_{S_t \cup S_t' \sim \mathcal{D}_t^{n_t+1}}\left(S_t \cup S_t' \text{ is bad}\right) \\
\leq&\frac{2\bar{\rho}_t^2}{\mu_t n_t} + \frac{4(\rho_t^2 - \bar{\rho}_t^2)\mathbb{E}_{S_t}\left[\varepsilon_t(\widehat{\mathbf{w}}_t)\right]}{\mu_t \bar{\varepsilon}_t n_t}. && \text{(equation (9))}
\end{aligned}$$

Since task $t-1$ and task $t$ satisfy $(r_t, \alpha)$ condition, there exists w', s.t. $\|w' - \widehat{w}_{t-1}\|_2 \leq r_t$ and $\varepsilon_t(w') \leq \alpha\varepsilon_{t-1}(\widehat{w}_{t-1})$ hold. Now we upper bound the excess risk of RERM:

$$
\begin{aligned}
\mathbb{E}_{S_t}\left[\varepsilon_t(\widehat{w}_t)\right] =& \mathbb{E}_{S_t \sim \mathcal{D}_t^{n_t}} L_{\mathcal{D}_t}(\widehat{w}_t) - \inf_{w} L_{\mathcal{D}_t}(w) \\
=& \mathbb{E}_{S_t \sim \mathcal{D}_t^{n_t}}\left(L_{\mathcal{D}_t}(\widehat{w}_t) - \widehat{L}_{S_t}(\widehat{w}_t)\right) + \mathbb{E}_{S_t \sim \mathcal{D}_t^{n_t}}\left(\widehat{L}_{S_t}(\widehat{w}_t) - \inf_{w} L_{\mathcal{D}_t}(w)\right) \\
\leq& \mathbb{E}_{S_t \sim \mathcal{D}_t^{n_t}}\left(L_{\mathcal{D}_t}(\widehat{w}_t) - \widehat{L}_{S_t}(\widehat{w}_t)\right) + \mathbb{E}_{S_t \sim \mathcal{D}_t^{n_t}}\left(\widehat{L}_{S_t}(\widehat{w}_t) + \frac{\mu_t}{2}\|\widehat{w}_t - \widehat{w}_{t-1}\|_2^2 - \inf_{w} L_{\mathcal{D}_t}(w)\right) \\
\leq& \mathbb{E}_{S_t \sim \mathcal{D}_t^{n_t}}\left(L_{\mathcal{D}_t}(\widehat{w}_t) - \widehat{L}_{S_t}(\widehat{w}_t)\right) + \mathbb{E}_{S_t \sim \mathcal{D}_t^{n_t}}\left(\widehat{L}_{S_t}(w') + \frac{\mu_t}{2}\|w' - \widehat{w}_{t-1}\|_2^2 - \inf_{w} L_{\mathcal{D}_t}(w)\right) \\
=& \mathbb{E}_{S_t \sim \mathcal{D}_t^{n_t}}\left(L_{\mathcal{D}_t}(\widehat{w}_t) - \widehat{L}_{S_t}(\widehat{w}_t)\right) + L_{\mathcal{D}_t}(w') + \frac{\mu_t}{2}\|w' - \widehat{w}_{t-1}\|_2^2 - \inf_{w} L_{\mathcal{D}_t}(w) \\
\leq& \frac{2\bar{\rho}_t^2}{\mu_t n_t} + \frac{4(\rho_t^2 - \bar{\rho}_t^2)\mathbb{E}_{S_t}\left[\varepsilon_t(\widehat{w}_t)\right]}{\mu_t \bar{\varepsilon}_t n_t} + \alpha\varepsilon_{t-1}(\widehat{w}_{t-1}) + \frac{\mu_t r_t^2}{2}.
\end{aligned}
$$

Taking expectation w.r.t. $S_1, \ldots, S_{t-1}$, we obtain

$$
\mathbb{E}\left[\varepsilon_t(\widehat{w}_t)\right] \leq \frac{2\bar{\rho}_t^2}{\mu_t n_t} + \frac{4(\rho_t^2 - \bar{\rho}_t^2)\mathbb{E}\left[\varepsilon_t(\widehat{w}_t)\right]}{\mu_t \bar{\varepsilon}_t n_t} + \alpha\mathbb{E}\left[\varepsilon_{t-1}(\widehat{w}_{t-1})\right] + \frac{\mu_t r_t^2}{2}.
$$

If $\frac{4(\rho_t^2 - \bar{\rho}_t^2)}{\mu_t \bar{\varepsilon}_t n_t} < 1$, we can solve the above inequality, and get

$$
\mathbb{E}\left[\varepsilon_t(\widehat{w}_t)\right] \leq \frac{\frac{2\bar{\rho}_t^2}{\mu_t n_t} + \alpha\mathbb{E}\left[\varepsilon_{t-1}(\widehat{w}_{t-1})\right] + \frac{\mu_t r_t^2}{2}}{1 - \frac{4(\rho_t^2 - \bar{\rho}_t^2)}{\mu_t \bar{\varepsilon}_t n_t}}.
$$

Denote $x = \sqrt{\frac{4\bar{\rho}_t^2}{n_t r_t^2} + \frac{32(\rho_t^2 - \bar{\rho}_t^2)^2}{(1-\alpha)\bar{\varepsilon}_t^2 n_t^2}} + \frac{4(\rho_t^2 - \bar{\rho}_t^2)}{\bar{\varepsilon}_t n_t}$, and select $\mu_t = \max\{\frac{4(1+\alpha)(\rho_t^2 - \bar{\rho}_t^2)}{(1-\alpha)\bar{\varepsilon}_t n_t}, x\}$. We get

$$
\begin{aligned}
\mathbb{E}\left[\varepsilon_t(\widehat{w}_t)\right] \leq& \frac{\frac{2\bar{\rho}_t^2}{\mu_t n_t} + \alpha\mathbb{E}\left[\varepsilon_{t-1}(\widehat{w}_{t-1})\right] + \frac{\mu_t r_t^2}{2}}{1 - \frac{4(\rho_t^2 - \bar{\rho}_t^2)}{\mu_t \bar{\varepsilon}_t n_t}} \\
=& \frac{\frac{2\bar{\rho}_t^2}{\mu_t n_t} + \frac{\mu_t r_t^2}{2}}{1 - \frac{4(\rho_t^2 - \bar{\rho}_t^2)}{\mu_t \bar{\varepsilon}_t n_t}} + \frac{\alpha\mathbb{E}\left[\varepsilon_{t-1}(\widehat{w}_{t-1})\right]}{1 - \frac{4(\rho_t^2 - \bar{\rho}_t^2)}{\mu_t \bar{\varepsilon}_t n_t}} \\
\leq& \frac{\frac{2\bar{\rho}_t^2}{x n_t} + \frac{r_t^2}{2}\mu_t}{1 - \frac{4(\rho_t^2 - \bar{\rho}_t^2)}{x \bar{\varepsilon}_t n_t}} + \frac{\alpha\mathbb{E}\left[\varepsilon_{t-1}(\widehat{w}_{t-1})\right]}{1 - \frac{1-\alpha}{1+\alpha}} \\
\leq& \frac{\frac{2\bar{\rho}_t^2}{x n_t} + \frac{r_t^2}{2}\left(x + \frac{4(1+\alpha)(\rho_t^2 - \bar{\rho}_t^2)}{(1-\alpha)\bar{\varepsilon}_t n_t}\right)}{1 - \frac{4(\rho_t^2 - \bar{\rho}_t^2)}{x \bar{\varepsilon}_t n_t}} + \frac{1+\alpha}{2}\mathbb{E}\left[\varepsilon_{t-1}(\widehat{w}_{t-1})\right] \\
=& \frac{\frac{2\bar{\rho}_t^2}{n_t} + \frac{r_t^2}{2}\left(x^2 + \frac{4(1+\alpha)(\rho_t^2 - \bar{\rho}_t^2)}{(1-\alpha)\bar{\varepsilon}_t n_t}x\right)}{x - \frac{4(\rho_t^2 - \bar{\rho}_t^2)}{\bar{\varepsilon}_t n_t}} + \frac{1+\alpha}{2}\mathbb{E}\left[\varepsilon_{t-1}(\widehat{w}_{t-1})\right] \\
=& r_t^2 \sqrt{\frac{4\bar{\rho}_t^2}{n_t r_t^2} + \frac{32(\rho_t^2 - \bar{\rho}_t^2)^2}{(1-\alpha)\bar{\varepsilon}_t^2 n_t^2}} + \frac{(6-2\alpha)r_t^2(\rho_t^2 - \bar{\rho}_t^2)}{(1-\alpha)\bar{\varepsilon}_t n_t} + \frac{1+\alpha}{2}\mathbb{E}\left[\varepsilon_{t-1}(\widehat{w}_{t-1})\right] \\
\leq& \frac{2r_t\bar{\rho}_t}{\sqrt{n_t}} + \left(\sqrt{\frac{32}{1-\alpha}} + \frac{6-2\alpha}{1-\alpha}\right)\frac{r_t^2(\rho_t^2 - \bar{\rho}_t^2)}{\bar{\varepsilon}_t n_t} + \frac{1+\alpha}{2}\mathbb{E}\left[\varepsilon_{t-1}(\widehat{w}_{t-1})\right] \\
& \hspace{6cm} (\sqrt{A+B} \leq \sqrt{A} + \sqrt{B}) \\
\leq& \frac{2r_t\bar{\rho}_t}{\sqrt{n_t}} + \frac{12}{1-\alpha}\frac{r_t^2(\rho_t^2 - \bar{\rho}_t^2)}{\bar{\varepsilon}_t n_t} + \frac{1+\alpha}{2}\mathbb{E}\left[\varepsilon_{t-1}(\widehat{w}_{t-1})\right] \\
=& \frac{2r_t}{\sqrt{n_t}}\left(\bar{\rho}_t + \frac{6r_t(\rho_t^2 - \bar{\rho}_t^2)}{(1-\alpha)\bar{\varepsilon}_t \sqrt{n_t}}\right) + \frac{1+\alpha}{2}\mathbb{E}\left[\varepsilon_{t-1}(\widehat{w}_{t-1})\right].
\end{aligned}
$$

$\square$

## B.4 Missing Proofs in Section 4.4

**Theorem 4.6.** Setting the regularization parameter $\mu_t = \max\{\frac{(2+6\alpha)H_t}{(1-\alpha)n_t}, \frac{1}{r_t}\sqrt{\frac{32H_tL_t^\star}{n_t}}\}$, we have

$$\mathbb{E}\left[\varepsilon_t(\widehat{w}_t)\right] \leq \sqrt{\frac{32L_t^\star H_t r_t^2}{n_t}} + \frac{9H_t r_t^2}{(1-\alpha)n_t} + \frac{1+\alpha}{2}\mathbb{E}\left[\varepsilon_{t-1}(\widehat{w}_{t-1})\right].$$

*Proof of Theorem 4.6.* Let $S_t = \{z_1, z_2, \ldots, z_{n_t}\} \sim \mathcal{D}_t^{n_t}$ and $S_t' = \{z_1', z_2, \ldots, z_{n_t}\} \sim \mathcal{D}_t^{n_t}$ be two neighboring data sets that differ in one single example. $S_t \cup S_t' = \{z_1', z_1, z_2, \ldots, z_{n_t}\} \sim \mathcal{D}_t^{n_t+1}$. Recall $\widehat{w}_t \in \underset{w}{\text{argmin}}\left(\widehat{L}_{S_t}(w) + \frac{\mu_t}{2}\|w - \widehat{w}_{t-1}\|_2^2\right); \widehat{w}_t' \in \underset{w}{\text{argmin}}\left(\widehat{L}_{S_t'}(w) + \frac{\mu_t}{2}\|w - \widehat{w}_{t-1}\|_2^2\right).$

Since the optimization objective $\widehat{L}_{S_t}(w) + \frac{\mu_t}{2}\|w - \widehat{w}_{t-1}\|_2^2$ is $\mu_t$-strongly convex, we have

$$\widehat{L}_{S_t}(\widehat{w}_t') + \frac{\mu_t}{2}\|\widehat{w}_t' - \widehat{w}_{t-1}\|_2^2 \geq \widehat{L}_{S_t}(\widehat{w}_t) + \frac{\mu_t}{2}\|\widehat{w}_t - \widehat{w}_{t-1}\|_2^2 + \frac{\mu_t}{2}\|\widehat{w}_t' - \widehat{w}_t\|_2^2.$$

Similarly,

$$\widehat{L}_{S_t'}(\widehat{w}_t) + \frac{\mu_t}{2}\|\widehat{w}_t - \widehat{w}_{t-1}\|_2^2 \geq \widehat{L}_{S_t'}(\widehat{w}_t') + \frac{\mu_t}{2}\|\widehat{w}_t' - \widehat{w}_{t-1}\|_2^2 + \frac{\mu_t}{2}\|\widehat{w}_t' - \widehat{w}_t\|_2^2.$$

Adding up these two inequalities,

$$\mu_t\|\widehat{w}_t' - \widehat{w}_t\|_2^2 \leq \frac{\ell_t(z_1; \widehat{w}_t') - \ell_t(z_1; \widehat{w}_t)}{n_t} + \frac{\ell_t(z_1'; \widehat{w}_t) - \ell_t(z_1'; \widehat{w}_t')}{n_t}. \tag{10}$$

By the smoothness assumption and using the self-bounded property,

$$\begin{aligned}
\ell_t(z_1; \widehat{w}_t') - \ell_t(z_1; \widehat{w}_t) &\leq \left\langle \nabla_w \ell_t(z_1; \widehat{w}_t), \widehat{w}_t' - \widehat{w}_t \right\rangle + \frac{H_t}{2}\|\widehat{w}_t' - \widehat{w}_t\|_2^2 \\
&\leq \|\nabla_w \ell_t(z_1; \widehat{w}_t)\|_2 \|\widehat{w}_t' - \widehat{w}_t\|_2 + \frac{H_t}{2}\|\widehat{w}_t' - \widehat{w}_t\|_2^2 \\
&\leq \sqrt{2H_t\ell_t(z_1; \widehat{w}_t)}\|\widehat{w}_t' - \widehat{w}_t\|_2 + \frac{H_t}{2}\|\widehat{w}_t' - \widehat{w}_t\|_2^2. 
\end{aligned} \tag{11}$$

Similarly,

$$\ell_t(z_1'; \widehat{w}_t) - \ell_t(z_1'; \widehat{w}_t') \leq \sqrt{2H_t\ell_t(z_1'; \widehat{w}_t')}\|\widehat{w}_t' - \widehat{w}_t\|_2 + \frac{H_t}{2}\|\widehat{w}_t' - \widehat{w}_t\|_2^2. \tag{12}$$

From the choice of $\mu_t$ we know that $\mu_t n_t > H_t$. Plugging these two inequalities into equation (10), we get

$$\|\widehat{w}_t' - \widehat{w}_t\|_2 \leq \frac{\sqrt{2H_t}}{\mu_t n_t - H_t}\left(\sqrt{\ell_t(z_1; \widehat{w}_t)} + \sqrt{\ell_t(z_1'; \widehat{w}_t')}\right).$$

Adding up equation (11) and equation (12), and combining with the inequality above, we get

$$\begin{aligned}
&\left(\ell_t(z_1; \widehat{w}_t') - \ell_t(z_1; \widehat{w}_t)\right) + \left(\ell_t(z_1'; \widehat{w}_t) - \ell_t(z_1'; \widehat{w}_t')\right) \\
&\leq \left(\frac{2H_t}{\mu_t n_t - H_t} + \frac{2H_t^2}{(\mu_t n_t - H_t)^2}\right)\left(\sqrt{\ell_t(z_1; \widehat{w}_t)} + \sqrt{\ell_t(z_1'; \widehat{w}_t')}\right)^2 \\
&\leq \left(\frac{4H_t}{\mu_t n_t - H_t} + \frac{4H_t^2}{(\mu_t n_t - H_t)^2}\right)\left(\ell_t(z_1; \widehat{w}_t) + \ell_t(z_1'; \widehat{w}_t')\right).
\end{aligned}$$

Now we upper bound the generalization gap of RERM:

$$\begin{aligned}
&\mathbb{E}_{S_t \sim \mathcal{D}_t^{n_t}}\left(L_{\mathcal{D}_t}(\widehat{w}_t) - \widehat{L}_{S_t}(\widehat{w}_t)\right) \\
&= \frac{1}{2}\mathbb{E}_{S_t \cup S_t' \sim \mathcal{D}_t^{n_t+1}}\left[\left(\ell_t(z_1; \widehat{w}_t') - \ell_t(z_1; \widehat{w}_t)\right) + \left(\ell_t(z_1'; \widehat{w}_t) - \ell_t(z_1'; \widehat{w}_t')\right)\right]
\end{aligned}$$

$$\leq \left( \frac{2H_t}{\mu_t n_t - H_t} + \frac{2H_t^2}{(\mu_t n_t - H_t)^2} \right) \mathbb{E}_{S_t \cup S_t' \sim \mathcal{D}_t^{n_t+1}} \left[ \ell_t(z_1; \widehat{w}_t) + \ell_t(z_1'; \widehat{w}_t') \right]$$

$$= \left( \frac{4H_t}{\mu_t n_t - H_t} + \frac{4H_t^2}{(\mu_t n_t - H_t)^2} \right) \mathbb{E}_{S_t \sim \mathcal{D}_t^{n_t}} \left[ \widehat{L}_{S_t}(\widehat{w}_t) \right].$$

Since task $t-1$ and task $t$ satisfy $(r_t, \alpha)$ condition, there exists w', s.t. $\|w' - \widehat{w}_{t-1}\|_2 \leq r_t$ and $\varepsilon_t(w') \leq \alpha \varepsilon_{t-1}(\widehat{w}_{t-1})$ hold. Now we upper bound the excess risk of RERM:

$$\mathbb{E}_{S_t} \left[ \varepsilon_t(\widehat{w}_t) \right]$$
$$= \mathbb{E}_{S_t \sim \mathcal{D}_t^{n_t}} L_{\mathcal{D}_t}(\widehat{w}_t) - L_t^\star$$

$$\leq \left( 1 + \frac{4H_t}{\mu_t n_t - H_t} + \frac{4H_t^2}{(\mu_t n_t - H_t)^2} \right) \mathbb{E}_{S_t \sim \mathcal{D}_t^{n_t}} \left[ \widehat{L}_{S_t}(\widehat{w}_t) \right] - L_t^\star$$

$$\leq \left( 1 + \frac{4H_t}{\mu_t n_t - H_t} + \frac{4H_t^2}{(\mu_t n_t - H_t)^2} \right) \mathbb{E}_{S_t \sim \mathcal{D}_t^{n_t}} \left[ \widehat{L}_{S_t}(\widehat{w}_t) + \frac{\mu_t}{2} \|\widehat{w}_t - \widehat{w}_{t-1}\|_2^2 \right] - L_t^\star$$

$$\leq \left( 1 + \frac{4H_t}{\mu_t n_t - H_t} + \frac{4H_t^2}{(\mu_t n_t - H_t)^2} \right) \mathbb{E}_{S_t \sim \mathcal{D}_t^{n_t}} \left[ \widehat{L}_{S_t}(w') + \frac{\mu_t}{2} \|w' - \widehat{w}_{t-1}\|_2^2 \right] - L_t^\star$$

$$\leq \left( 1 + \frac{4H_t}{\mu_t n_t - H_t} + \frac{4H_t^2}{(\mu_t n_t - H_t)^2} \right) \left( L_{\mathcal{D}_t}(w') + \frac{\mu_t}{2} \|w' - \widehat{w}_{t-1}\|_2^2 \right) - L_t^\star$$

$$\leq \left( 1 + \frac{4H_t}{\mu_t n_t - H_t} + \frac{4H_t^2}{(\mu_t n_t - H_t)^2} \right) \left( L_t^\star + \alpha \varepsilon_{t-1}(\widehat{w}_{t-1}) + \frac{\mu_t}{2} r_t^2 \right) - L_t^\star$$

$$= \left( 1 + \frac{4H_t}{\mu_t n_t - H_t} + \frac{4H_t^2}{(\mu_t n_t - H_t)^2} \right) \alpha \varepsilon_{t-1}(\widehat{w}_{t-1})$$

$$+ \left( 1 + \frac{4H_t}{\mu_t n_t - H_t} + \frac{4H_t^2}{(\mu_t n_t - H_t)^2} \right) \left( L_t^\star + \frac{\mu_t}{2} r_t^2 \right) - L_t^\star. \tag{13}$$

Since $mu_t \geq \frac{(2+6\alpha)H_t}{(1-\alpha)n_t}$,

$$\left( 1 + \frac{4H_t}{\mu_t n_t - H_t} + \frac{4H_t^2}{(\mu_t n_t - H_t)^2} \right) \alpha \leq \left( 1 + \frac{4(1-\alpha)}{1+7\alpha} + \frac{4(1-\alpha)^2}{(1+7\alpha)^2} \right) \alpha$$

$$= \left( 1 + \frac{8+24\alpha}{(1+7\alpha)^2}(1-\alpha) \right) \alpha$$

$$\leq \left( 1 + \frac{1-\alpha}{2\alpha} \right) \alpha = \frac{1+\alpha}{2};$$

$$\left( 1 + \frac{4H_t}{\mu_t n_t - H_t} + \frac{4H_t^2}{(\mu_t n_t - H_t)^2} \right) \leq \left( 1 + \frac{8H_t}{\mu_t n_t} + \frac{8H_t}{\mu_t n_t} \right)$$

$$= \left( 1 + \frac{16H_t}{\mu_t n_t} \right).$$

Plugging these two inequalities into equation (13), we get

$$\mathbb{E}_{S_t} \left[ \varepsilon_t(\widehat{w}_t) \right]$$
$$\leq \frac{1+\alpha}{2} \varepsilon_{t-1}(\widehat{w}_{t-1}) + \left( 1 + \frac{16H_t}{\mu_t n_t} \right) \left( L_t^\star + \frac{\mu_t}{2} r_t^2 \right) - L_t^\star$$

$$= \frac{1+\alpha}{2} \varepsilon_{t-1}(\widehat{w}_{t-1}) + \frac{8H_t r_t^2}{n_t} + \frac{r_t^2}{2} \mu_t + 16 L_t^\star \frac{H_t}{\mu_t n_t}$$

$$\leq \frac{1+\alpha}{2}\varepsilon_{t-1}(\widehat{w}_{t-1}) + \frac{8H_t r_t^2}{n_t} + \frac{r_t^2}{2}\left(\frac{(2+6\alpha)H_t}{(1-\alpha)n_t} + \frac{1}{r_t}\sqrt{\frac{32H_t L_t^\star}{n_t}}\right) + 16L_t^\star \frac{H_t}{\left(\frac{1}{r_t}\sqrt{\frac{32H_t L_t^\star}{n_t}}\right)n_t}$$

$$= \frac{1+\alpha}{2}\varepsilon_{t-1}(\widehat{w}_{t-1}) + \frac{(9-5\alpha)H_t r_t^2}{(1-\alpha)n_t} + r_t\sqrt{\frac{32H_t L_t^\star}{n_t}}$$

$$\leq \sqrt{\frac{32L_t^\star H_t r_t^2}{n_t}} + \frac{9H_t r_t^2}{(1-\alpha)n_t} + \frac{1+\alpha}{2}\varepsilon_{t-1}(\widehat{w}_{t-1}).$$

Taking expectation w.r.t. $S_1, \ldots, S_{t-1}$, we obtain

$$\mathbb{E}\left[\varepsilon_t(\widehat{w}_t)\right] \leq \sqrt{\frac{32L_t^\star H_t r_t^2}{n_t}} + \frac{9H_t r_t^2}{(1-\alpha)n_t} + \frac{1+\alpha}{2}\mathbb{E}\left[\varepsilon_{t-1}(\widehat{w}_{t-1})\right].$$

$\square$

## B.5  Missing Proofs for Adversarial Robustness

**Proposition B.1.** *If the standard loss $\ell_t((x,y);w)$ is convex, then the robust loss $\ell_t^{rob}((x,y);w) := \sup_{\tilde{x}\in\mathcal{B}(x)} \ell_t((\tilde{x},y);w)$ is convex.*

*Proof of Proposition B.1.* $\forall w_1, w_2, \lambda \in [0,1]$,

$$\ell_t^{rob}((x,y);\lambda w_1 + (1-\lambda)w_2) = \sup_{\tilde{x}\in\mathcal{B}(x)} \ell_t((\tilde{x},y);\lambda w_1 + (1-\lambda)w_2)$$
$$\leq \sup_{\tilde{x}\in\mathcal{B}(x)} [\lambda\ell_t((\tilde{x},y);w_1) + (1-\lambda)\ell_t((\tilde{x},y);w_2)]$$
$$\leq \sup_{\tilde{x}\in\mathcal{B}(x)} [\lambda\ell_t((\tilde{x},y);w_1)] + \sup_{\tilde{x}\in\mathcal{B}(x)} [(1-\lambda)\ell_t((\tilde{x},y);w_2)]$$
$$= \lambda\ell_t^{rob}((x,y);w_1) + (1-\lambda)\ell_t^{rob}((x,y);w_2).$$

$\square$

**Proposition B.2.** *If the standard loss $\ell_t((x,y);w)$ is $\rho_t$-Lipschitz, then the robust loss $\ell_t^{rob}((x,y);w) := \sup_{\tilde{x}\in\mathcal{B}(x)} \ell_t((\tilde{x},y);w)$ is $\rho_t$-Lipschitz.*

*Proof of Proposition B.1.* $\forall w_1, w_2$,

$$\ell_t^{rob}((x,y);w_1) - \ell_t^{rob}((x,y);w_2) = \sup_{\tilde{x}\in\mathcal{B}(x)} \ell_t((\tilde{x},y);w_1) - \sup_{\tilde{x}\in\mathcal{B}(x)} \ell_t((\tilde{x},y);w_2)$$
$$\leq \sup_{\tilde{x}\in\mathcal{B}(x)} [\ell_t((\tilde{x},y);w_1) - \ell_t((\tilde{x},y);w_2)]$$
$$\leq \sup_{\tilde{x}\in\mathcal{B}(x)} (\rho_t\|w_1 - w_2\|_2)$$
$$= \rho_t\|w_1 - w_2\|_2.$$

$\square$

Now we prove Theorem 4.6 in the adversarial robustness setting. In this setting, all tasks are learning the robust loss; the empirical risk, expected risk and excess risk are defined using the robust loss. We assume that $\forall z$, the standard loss $\ell_t(z;w)$ is convex, $H_t$-smooth and nonnegative. In addition, task $t-1$ and task $t$ satisfy $(r_t, \alpha)$ condition for constants $r_t > 0$ and $\alpha \in (0,1)$. Denote $L_t^\star = \inf_w L_{\mathcal{D}_t}^{rob}(w)$. We use biased RERM described in Algorithm 1 to learn these tasks. We focus on two consecutive tasks: task $t-1$ and task $t$ as before.

**Theorem B.3.** Setting the regularization parameter $\mu_t = \max\{\frac{(2+6\alpha)H_t}{(1-\alpha)n_t}, \frac{1}{r_t}\sqrt{\frac{32H_t L_t^\star}{n_t}}\}$, we have

$$\mathbb{E}\left[\varepsilon_t(\widehat{w}_t)\right] \leq \sqrt{\frac{32L_t^\star H_t r_t^2}{n_t}} + \frac{9H_t r_t^2}{(1-\alpha)n_t} + \frac{1+\alpha}{2}\mathbb{E}\left[\varepsilon_{t-1}(\widehat{w}_{t-1})\right].$$

*Proof of Theorem B.3.* Let $S_t = \{z_1, z_2, \ldots, z_{n_t}\} \sim \mathcal{D}_t^{n_t}$ and $S_t' = \{z_1', z_2, \ldots, z_{n_t}\} \sim \mathcal{D}_t^{n_t}$ be two neighboring data sets that differ in one single example. $S_t \cup S_t' = \{z_1', z_1, z_2, \ldots, z_{n_t}\} \sim \mathcal{D}_t^{n_t+1}$. Recall $\widehat{w}_t \in \operatorname*{argmin}_{w}\left(\widehat{L}_{S_t}^{rob}(w) + \frac{\mu_t}{2}\|w - \widehat{w}_{t-1}\|_2^2\right)$; $\widehat{w}_t' \in \operatorname*{argmin}_{w}\left(\widehat{L}_{S_t'}^{rob}(w) + \frac{\mu_t}{2}\|w - \widehat{w}_{t-1}\|_2^2\right)$.

Since the optimization objective $\widehat{L}_{S_t}^{rob}(w) + \frac{\mu_t}{2}\|w - \widehat{w}_{t-1}\|_2^2$ is $\mu_t$-strongly convex, we have

$$\widehat{L}_{S_t}^{rob}(\widehat{w}_t') + \frac{\mu_t}{2}\|\widehat{w}_t' - \widehat{w}_{t-1}\|_2^2 \geq \widehat{L}_{S_t}^{rob}(\widehat{w}_t) + \frac{\mu_t}{2}\|\widehat{w}_t - \widehat{w}_{t-1}\|_2^2 + \frac{\mu_t}{2}\|\widehat{w}_t' - \widehat{w}_t\|_2^2.$$

Similarly,

$$\widehat{L}_{S_t'}^{rob}(\widehat{w}_t) + \frac{\mu_t}{2}\|\widehat{w}_t - \widehat{w}_{t-1}\|_2^2 \geq \widehat{L}_{S_t'}^{rob}(\widehat{w}_t') + \frac{\mu_t}{2}\|\widehat{w}_t' - \widehat{w}_{t-1}\|_2^2 + \frac{\mu_t}{2}\|\widehat{w}_t' - \widehat{w}_t\|_2^2.$$

Adding up these two inequalities,

$$\mu_t\|\widehat{w}_t' - \widehat{w}_t\|_2^2 \leq \frac{\ell_t^{rob}(z_1; \widehat{w}_t') - \ell_t^{rob}(z_1; \widehat{w}_t)}{n_t} + \frac{\ell_t^{rob}(z_1'; \widehat{w}_t) - \ell_t^{rob}(z_1'; \widehat{w}_t')}{n_t}. \tag{14}$$

By the smoothness assumption and using the self-bounded property for the standard loss,

$$
\begin{aligned}
\ell_t^{rob}(z_1; \widehat{w}_t') - \ell_t^{rob}(z_1; \widehat{w}_t) &= \sup_{\tilde{x} \in \mathcal{B}(x_1)} \ell_t((\tilde{x}, y_1); \widehat{w}_t') - \sup_{\tilde{x} \in \mathcal{B}(x_1)} \ell_t((\tilde{x}, y_1); \widehat{w}_t) \\
&\leq \sup_{\tilde{x} \in \mathcal{B}(x_1)} \left[\ell_t((\tilde{x}, y_1); \widehat{w}_t') - \ell_t((\tilde{x}, y_1); \widehat{w}_t)\right] \\
&\leq \sup_{\tilde{x} \in \mathcal{B}(x_1)} \left\langle \nabla_w \ell_t((\tilde{x}, y_1); \widehat{w}_t), \widehat{w}_t' - \widehat{w}_t \right\rangle + \frac{H_t}{2}\|\widehat{w}_t' - \widehat{w}_t\|_2^2 \\
&\leq \sup_{\tilde{x} \in \mathcal{B}(x_1)} \|\nabla_w \ell_t((\tilde{x}, y_1); \widehat{w}_t)\|_2 \|\widehat{w}_t' - \widehat{w}_t\|_2 + \frac{H_t}{2}\|\widehat{w}_t' - \widehat{w}_t\|_2^2 \\
&\leq \sup_{\tilde{x} \in \mathcal{B}(x_1)} \sqrt{2H_t \ell_t((\tilde{x}, y_1); \widehat{w}_t)} \|\widehat{w}_t' - \widehat{w}_t\|_2 + \frac{H_t}{2}\|\widehat{w}_t' - \widehat{w}_t\|_2^2 \\
&= \sqrt{2H_t \ell_t^{rob}(z_1; \widehat{w}_t)} \|\widehat{w}_t' - \widehat{w}_t\|_2 + \frac{H_t}{2}\|\widehat{w}_t' - \widehat{w}_t\|_2^2. \tag{15}
\end{aligned}
$$

Similarly,

$$\ell_t^{rob}(z_1'; \widehat{w}_t) - \ell_t^{rob}(z_1'; \widehat{w}_t') \leq \sqrt{2H_t \ell_t^{rob}(z_1'; \widehat{w}_t')} \|\widehat{w}_t' - \widehat{w}_t\|_2 + \frac{H_t}{2}\|\widehat{w}_t' - \widehat{w}_t\|_2^2. \tag{16}$$

From the choice of $\mu_t$ we know that $\mu_t n_t > H_t$. Plugging these two inequalities into equation (14), we get

$$\|\widehat{w}_t' - \widehat{w}_t\|_2 \leq \frac{\sqrt{2H_t}}{\mu_t n_t - H_t}\left(\sqrt{\ell_t^{rob}(z_1; \widehat{w}_t)} + \sqrt{\ell_t^{rob}(z_1'; \widehat{w}_t')}\right).$$

Adding up equation (15) and equation (16), and combining with the inequality above, we get

$$
\begin{aligned}
&\left(\ell_t^{rob}(z_1; \widehat{w}_t') - \ell_t^{rob}(z_1; \widehat{w}_t)\right) + \left(\ell_t^{rob}(z_1'; \widehat{w}_t) - \ell_t^{rob}(z_1'; \widehat{w}_t')\right) \\
&\leq \left(\frac{2H_t}{\mu_t n_t - H_t} + \frac{2H_t^2}{(\mu_t n_t - H_t)^2}\right)\left(\sqrt{\ell_t^{rob}(z_1; \widehat{w}_t)} + \sqrt{\ell_t^{rob}(z_1'; \widehat{w}_t')}\right)^2 \\
&\leq \left(\frac{4H_t}{\mu_t n_t - H_t} + \frac{4H_t^2}{(\mu_t n_t - H_t)^2}\right)\left(\ell_t^{rob}(z_1; \widehat{w}_t) + \ell_t^{rob}(z_1'; \widehat{w}_t')\right).
\end{aligned}
$$

Now we upper bound the generalization gap of RERM:

$$\mathbb{E}_{S_t \sim \mathcal{D}_t^{n_t}} \left( L_{\mathcal{D}_t}^{rob}(\widehat{w}_t) - \widehat{L}_{S_t}^{rob}(\widehat{w}_t) \right)$$

$$= \frac{1}{2} \mathbb{E}_{S_t \cup S_t' \sim \mathcal{D}_t^{n_t+1}} \left[ \left( \ell_t^{rob}(z_1; \widehat{w}_t') - \ell_t^{rob}(z_1; \widehat{w}_t) \right) + \left( \ell_t^{rob}(z_1'; \widehat{w}_t) - \ell_t^{rob}(z_1'; \widehat{w}_t') \right) \right]$$

$$\leq \left( \frac{2H_t}{\mu_t n_t - H_t} + \frac{2H_t^2}{(\mu_t n_t - H_t)^2} \right) \mathbb{E}_{S_t \cup S_t' \sim \mathcal{D}_t^{n_t+1}} \left[ \ell_t^{rob}(z_1; \widehat{w}_t) + \ell_t^{rob}(z_1'; \widehat{w}_t') \right]$$

$$= \left( \frac{4H_t}{\mu_t n_t - H_t} + \frac{4H_t^2}{(\mu_t n_t - H_t)^2} \right) \mathbb{E}_{S_t \sim \mathcal{D}_t^{n_t}} \left[ \widehat{L}_{S_t}^{rob}(\widehat{w}_t) \right].$$

Since task $t-1$ and task $t$ satisfy $(r_t, \alpha)$ condition, there exists w', s.t. $\|w' - \widehat{w}_{t-1}\|_2 \leq r_t$ and $\varepsilon_t(w') \leq \alpha \varepsilon_{t-1}(\widehat{w}_{t-1})$ hold. Now we upper bound the excess risk of RERM:

$$\mathbb{E}_{S_t} \left[ \varepsilon_t(\widehat{w}_t) \right]$$

$$= \mathbb{E}_{S_t \sim \mathcal{D}_t^{n_t}} L_{\mathcal{D}_t}^{rob}(\widehat{w}_t) - L_t^{\star}$$

$$\leq \left( 1 + \frac{4H_t}{\mu_t n_t - H_t} + \frac{4H_t^2}{(\mu_t n_t - H_t)^2} \right) \mathbb{E}_{S_t \sim \mathcal{D}_t^{n_t}} \left[ \widehat{L}_{S_t}^{rob}(\widehat{w}_t) \right] - L_t^{\star}$$

$$\leq \left( 1 + \frac{4H_t}{\mu_t n_t - H_t} + \frac{4H_t^2}{(\mu_t n_t - H_t)^2} \right) \mathbb{E}_{S_t \sim \mathcal{D}_t^{n_t}} \left[ \widehat{L}_{S_t}^{rob}(\widehat{w}_t) + \frac{\mu_t}{2} \|\widehat{w}_t - \widehat{w}_{t-1}\|_2^2 \right] - L_t^{\star}$$

$$\leq \left( 1 + \frac{4H_t}{\mu_t n_t - H_t} + \frac{4H_t^2}{(\mu_t n_t - H_t)^2} \right) \mathbb{E}_{S_t \sim \mathcal{D}_t^{n_t}} \left[ \widehat{L}_{S_t}^{rob}(w') + \frac{\mu_t}{2} \|w' - \widehat{w}_{t-1}\|_2^2 \right] - L_t^{\star}$$

$$\leq \left( 1 + \frac{4H_t}{\mu_t n_t - H_t} + \frac{4H_t^2}{(\mu_t n_t - H_t)^2} \right) \left( L_{\mathcal{D}_t}^{rob}(w') + \frac{\mu_t}{2} \|w' - \widehat{w}_{t-1}\|_2^2 \right) - L_t^{\star}$$

$$\leq \left( 1 + \frac{4H_t}{\mu_t n_t - H_t} + \frac{4H_t^2}{(\mu_t n_t - H_t)^2} \right) \left( L_t^{\star} + \alpha \varepsilon_{t-1}(\widehat{w}_{t-1}) + \frac{\mu_t}{2} r_t^2 \right) - L_t^{\star}$$

$$= \left( 1 + \frac{4H_t}{\mu_t n_t - H_t} + \frac{4H_t^2}{(\mu_t n_t - H_t)^2} \right) \alpha \varepsilon_{t-1}(\widehat{w}_{t-1})$$

$$+ \left( 1 + \frac{4H_t}{\mu_t n_t - H_t} + \frac{4H_t^2}{(\mu_t n_t - H_t)^2} \right) \left( L_t^{\star} + \frac{\mu_t}{2} r_t^2 \right) - L_t^{\star}. \tag{17}$$

Since $\mu_t \geq \frac{(2+6\alpha)H_t}{(1-\alpha)n_t}$,

$$\left( 1 + \frac{4H_t}{\mu_t n_t - H_t} + \frac{4H_t^2}{(\mu_t n_t - H_t)^2} \right) \alpha \leq \left( 1 + \frac{4(1-\alpha)}{1+7\alpha} + \frac{4(1-\alpha)^2}{(1+7\alpha)^2} \right) \alpha$$

$$= \left( 1 + \frac{8+24\alpha}{(1+7\alpha)^2}(1-\alpha) \right) \alpha$$

$$\leq \left( 1 + \frac{1-\alpha}{2\alpha} \right) \alpha = \frac{1+\alpha}{2};$$

$$\left( 1 + \frac{4H_t}{\mu_t n_t - H_t} + \frac{4H_t^2}{(\mu_t n_t - H_t)^2} \right) \leq \left( 1 + \frac{8H_t}{\mu_t n_t} + \frac{8H_t}{\mu_t n_t} \right)$$

$$= \left( 1 + \frac{16H_t}{\mu_t n_t} \right).$$

Plugging these two inequalities into equation (17), we get

$$\mathbb{E}_{S_t} \left[ \varepsilon_t(\widehat{w}_t) \right]$$

$$\leq \frac{1+\alpha}{2}\varepsilon_{t-1}(\widehat{w}_{t-1}) + \left(1 + \frac{16H_t}{\mu_t n_t}\right)\left(L_t^\star + \frac{\mu_t}{2}r_t^2\right) - L_t^\star$$

$$= \frac{1+\alpha}{2}\varepsilon_{t-1}(\widehat{w}_{t-1}) + \frac{8H_t r_t^2}{n_t} + \frac{r_t^2}{2}\mu_t + 16L_t^\star\frac{H_t}{\mu_t n_t}$$

$$\leq \frac{1+\alpha}{2}\varepsilon_{t-1}(\widehat{w}_{t-1}) + \frac{8H_t r_t^2}{n_t} + \frac{r_t^2}{2}\left(\frac{(2+6\alpha)H_t}{(1-\alpha)n_t} + \frac{1}{r_t}\sqrt{\frac{32H_t L_t^\star}{n_t}}\right) + 16L_t^\star\frac{H_t}{\left(\frac{1}{r_t}\sqrt{\frac{32H_t L_t^\star}{n_t}}\right)n_t}$$

$$= \frac{1+\alpha}{2}\varepsilon_{t-1}(\widehat{w}_{t-1}) + \frac{(9-5\alpha)H_t r_t^2}{(1-\alpha)n_t} + r_t\sqrt{\frac{32H_t L_t^\star}{n_t}}$$

$$\leq \sqrt{\frac{32L_t^\star H_t r_t^2}{n_t}} + \frac{9H_t r_t^2}{(1-\alpha)n_t} + \frac{1+\alpha}{2}\varepsilon_{t-1}(\widehat{w}_{t-1}).$$

Taking expectation w.r.t. $S_1, \ldots, S_{t-1}$, we obtain

$$\mathbb{E}\left[\varepsilon_t(\widehat{w}_t)\right] \leq \sqrt{\frac{32L_t^\star H_t r_t^2}{n_t}} + \frac{9H_t r_t^2}{(1-\alpha)n_t} + \frac{1+\alpha}{2}\mathbb{E}\left[\varepsilon_{t-1}(\widehat{w}_{t-1})\right].$$

$\square$

## C  Missing Proofs in Section 5

**Lemma 5.1.** Let $\delta \in (0,1)$ and $\epsilon > 0$. If $n_t \geq \frac{8r_t^2\rho_t^2}{\epsilon^2}\left(\ln\left(\frac{2}{\delta}\right) + m\ln\left(\frac{8r_t\rho_t}{\epsilon}+1\right)\right)$, then with probability at least $1-\delta$ over the randomness of $S_t$,

$$\sup_{w:\|w-\widehat{w}_{t-1}\|_2 \leq r_t} |\widehat{L}_{S_t}(w) - L_{\mathcal{D}_t}(w) - \widehat{L}_{S_t}(\widehat{w}_{t-1}) + L_{\mathcal{D}_t}(\widehat{w}_{t-1})| \leq \epsilon.$$

*Proof of Lemma 5.1.* Define $f(z;w) = \ell_t(z;w) - \ell_t(z;\widehat{w}_{t-1})$, then

$$\widehat{L}_{S_t}(w) - L_{\mathcal{D}_t}(w) - \widehat{L}_{S_t}(\widehat{w}_{t-1}) + L_{\mathcal{D}_t}(\widehat{w}_{t-1}) = \frac{1}{n_t}\sum_{i=1}^{n_t} f(z_i;w) - \mathbb{E}_{z\sim\mathcal{D}_t}f(z;w).$$

From the Lipschitz assumption, if $\|w-\widehat{w}_{t-1}\|_2 \leq r_t$, $|f(z;w)| \leq r_t\rho_t$ is bounded. From Vershynin [2018] Chapter 4, let $\{v_1,\ldots,v_K\}$ be an $\frac{\epsilon}{4\rho_t}$-net of $\{w : \|w-\widehat{w}_{t-1}\|_2 \leq r_t\}$, such that $K \leq \left(\frac{8r_t\rho_t}{\epsilon}+1\right)^m$. $\forall v_j$, from Hoeffding's Inequality, we get

$$\mathbb{P}_{S_t}\left(\left|\frac{1}{n_t}\sum_{i=1}^{n_t} f(z_i;v_j) - \mathbb{E}_{z\sim\mathcal{D}_t}f(z;v_j)\right| > \frac{\epsilon}{2}\right) \leq 2\exp\left(-\frac{\epsilon^2}{8r_t^2\rho_t^2}n_t\right).$$

Taking a union bound,

$$\mathbb{P}_{S_t}\left(\left|\frac{1}{n_t}\sum_{i=1}^{n_t} f(z_i;v_j) - \mathbb{E}_{z\sim\mathcal{D}_t}f(z;v_j)\right| \leq \frac{\epsilon}{2}, \forall j\right) \geq 1 - 2K\exp\left(-\frac{\epsilon^2}{8r_t^2\rho_t^2}n_t\right)$$

$$\geq 1 - 2\left(\frac{8r_t\rho_t}{\epsilon}+1\right)^m \exp\left(-\frac{\epsilon^2}{8r_t^2\rho_t^2}n_t\right)$$

$$\geq 1 - \delta.$$

If the event $\left|\frac{1}{n_t}\sum_{i=1}^{n_t} f(z_i;v_j) - \mathbb{E}_{z\sim\mathcal{D}_t}f(z;v_j)\right| \leq \frac{\epsilon}{2}$ holds for all $v_j$, I claim

$$\sup_{w:\|w-\widehat{w}_{t-1}\|_2 \leq r_t} |\widehat{L}_{S_t}(w) - L_{\mathcal{D}_t}(w) - \widehat{L}_{S_t}(\widehat{w}_{t-1}) + L_{\mathcal{D}_t}(\widehat{w}_{t-1})| \leq \epsilon.$$

$\forall \mathrm{w}$ that satisfies $\|\mathrm{w} - \widehat{\mathrm{w}}_{t-1}\|_2 \leq r_t$, from the definition of the net, there exists $\mathrm{v}_j$ such that $\|\mathrm{w} - \mathrm{v}_j\|_2 \leq \frac{\epsilon}{4\rho_t}$. Using triangle inequality,

$$
\begin{aligned}
&|\widehat{L}_{S_t}(\mathrm{w}) - L_{\mathcal{D}_t}(\mathrm{w}) - \widehat{L}_{S_t}(\widehat{\mathrm{w}}_{t-1}) + L_{\mathcal{D}_t}(\widehat{\mathrm{w}}_{t-1})| \\
\leq & |\widehat{L}_{S_t}(\mathrm{v}_j) - L_{\mathcal{D}_t}(\mathrm{v}_j) - \widehat{L}_{S_t}(\widehat{\mathrm{w}}_{t-1}) + L_{\mathcal{D}_t}(\widehat{\mathrm{w}}_{t-1})| + |\widehat{L}_{S_t}(\mathrm{v}_j) - \widehat{L}_{S_t}(\mathrm{w})| + |L_{\mathcal{D}_t}(\mathrm{v}_j) - L_{\mathcal{D}_t}(\mathrm{w})| \\
= & \left| \frac{1}{n_t} \sum_{i=1}^{n_t} f(\mathrm{z}_i; \mathrm{v}_j) - \mathbb{E}_{\mathrm{z} \sim \mathcal{D}_t} f(\mathrm{z}; \mathrm{v}_j) \right| + |\widehat{L}_{S_t}(\mathrm{v}_j) - \widehat{L}_{S_t}(\mathrm{w})| + |L_{\mathcal{D}_t}(\mathrm{v}_j) - L_{\mathcal{D}_t}(\mathrm{w})| \\
\leq & \frac{\epsilon}{2} + 2\rho_t \|\mathrm{w} - \mathrm{v}_j\|_2 \leq \frac{\epsilon}{2} + 2\rho_t \frac{\epsilon}{4\rho_t} = \epsilon.
\end{aligned}
$$

Therefore,

$$
\begin{aligned}
\mathbb{P}_{S_t} & \left( \sup_{\mathrm{w}: \|\mathrm{w} - \widehat{\mathrm{w}}_{t-1}\|_2 \leq r_t} |\widehat{L}_{S_t}(\mathrm{w}) - L_{\mathcal{D}_t}(\mathrm{w}) - \widehat{L}_{S_t}(\widehat{\mathrm{w}}_{t-1}) + L_{\mathcal{D}_t}(\widehat{\mathrm{w}}_{t-1})| \leq \epsilon \right) \\
\geq & \mathbb{P}_{S_t} \left( \left| \frac{1}{n_t} \sum_{i=1}^{n_t} f(\mathrm{z}_i; \mathrm{v}_j) - \mathbb{E}_{\mathrm{z} \sim \mathcal{D}_t} f(\mathrm{z}; \mathrm{v}_j) \right| \leq \frac{\epsilon}{2}, \forall j \right) \\
\geq & 1 - \delta.
\end{aligned}
$$

$\square$

**Theorem 5.2.** For any $\epsilon > 0$, if $n_t \geq \frac{8r_t^2 \rho_t^2}{\epsilon^2} \left( \ln\left(\frac{2}{\delta}\right) + m \ln\left(\frac{8r_t \rho_t}{\epsilon} + 1\right) \right)$, then with probability at least $1 - \delta$ over the randomness of $S_t$, we have $\varepsilon_t(\widehat{\mathrm{w}}_t) \leq 2\epsilon + \alpha \varepsilon_{t-1}(\widehat{\mathrm{w}}_{t-1})$.

*Proof of Theorem 5.2.* From Lemma 5.1, with probability at least $1 - \delta$,

$$
\sup_{\mathrm{w}: \|\mathrm{w} - \widehat{\mathrm{w}}_{t-1}\|_2 \leq r_t} |\widehat{L}_{S_t}(\mathrm{w}) - L_{\mathcal{D}_t}(\mathrm{w}) - \widehat{L}_{S_t}(\widehat{\mathrm{w}}_{t-1}) + L_{\mathcal{D}_t}(\widehat{\mathrm{w}}_{t-1})| \leq \epsilon.
$$

Since task $t - 1$ and task $t$ satisfy $(r_t, \alpha)$ condition, there exists $\mathrm{w}'$, s.t. $\|\mathrm{w}' - \widehat{\mathrm{w}}_{t-1}\|_2 \leq r_t$ and $\varepsilon_t(\mathrm{w}') \leq \alpha \varepsilon_{t-1}(\widehat{\mathrm{w}}_{t-1})$ hold. Now we upper bound the excess risk:

$$
\begin{aligned}
\varepsilon_t(\widehat{\mathrm{w}}_t) = & L_{\mathcal{D}_t}(\widehat{\mathrm{w}}_t) - \inf_{\mathrm{w}} L_{\mathcal{D}_t}(\mathrm{w}) \\
= & \left( L_{\mathcal{D}_t}(\widehat{\mathrm{w}}_t) - \widehat{L}_{S_t}(\widehat{\mathrm{w}}_t) \right) + \left( \widehat{L}_{S_t}(\widehat{\mathrm{w}}_t) - \inf_{\mathrm{w}} L_{\mathcal{D}_t}(\mathrm{w}) \right) \\
= & \left( L_{\mathcal{D}_t}(\widehat{\mathrm{w}}_t) - \widehat{L}_{S_t}(\widehat{\mathrm{w}}_t) - L_{\mathcal{D}_t}(\widehat{\mathrm{w}}_{t-1}) + \widehat{L}_{S_t}(\widehat{\mathrm{w}}_{t-1}) \right) \\
& \quad\quad + \left( L_{\mathcal{D}_t}(\widehat{\mathrm{w}}_{t-1}) - \widehat{L}_{S_t}(\widehat{\mathrm{w}}_{t-1}) \right) + \left( \widehat{L}_{S_t}(\widehat{\mathrm{w}}_t) - \inf_{\mathrm{w}} L_{\mathcal{D}_t}(\mathrm{w}) \right) \\
\leq & \epsilon + \left( L_{\mathcal{D}_t}(\widehat{\mathrm{w}}_{t-1}) - \widehat{L}_{S_t}(\widehat{\mathrm{w}}_{t-1}) \right) + \left( \widehat{L}_{S_t}(\mathrm{w}') - \inf_{\mathrm{w}} L_{\mathcal{D}_t}(\mathrm{w}) \right) \\
= & \epsilon + \left( \widehat{L}_{S_t}(\mathrm{w}') - L_{\mathcal{D}_t}(\mathrm{w}') - \widehat{L}_{S_t}(\widehat{\mathrm{w}}_{t-1}) + L_{\mathcal{D}_t}(\widehat{\mathrm{w}}_{t-1}) \right) + \left( L_{\mathcal{D}_t}(\mathrm{w}') - \inf_{\mathrm{w}} L_{\mathcal{D}_t}(\mathrm{w}) \right) \\
\leq & 2\epsilon + \alpha \varepsilon_{t-1}(\widehat{\mathrm{w}}_{t-1}).
\end{aligned}
$$

$\square$

**Corollary 5.3.** Assume $\varepsilon_1(\widehat{\mathrm{w}}_1) \leq \epsilon$. If $n_t \geq \frac{32r_t^2 \rho_t^2}{(1-\alpha)^2 \epsilon^2} \left( \ln\left(\frac{2T}{\delta}\right) + m \ln\left(\frac{16r_t \rho_t}{(1-\alpha)\epsilon} + 1\right) \right)$, for all $t \in 2, \ldots, T$, then Algorithm 4 ensures that with probability at least $1 - \delta$, we have that $\varepsilon_T(\widehat{\mathrm{w}}_T) \leq \epsilon$.

*Proof of Corollary 5.3.* Replacing $\epsilon$ with $\frac{1-\alpha}{2}\epsilon$ in Theorem 5.2, we know that $\varepsilon_t(\widehat{\mathrm{w}}_t) \leq (1 - \alpha)\epsilon + \alpha \varepsilon_{t-1}(\widehat{\mathrm{w}}_{t-1})$ holds with probability at least $1 - \frac{\delta}{T}$. Taking a union bound, with probability at least $1 - \delta$, $\varepsilon_t(\widehat{\mathrm{w}}_t) \leq (1 - \alpha)\epsilon + \alpha \varepsilon_{t-1}(\widehat{\mathrm{w}}_{t-1})$ holds for every $t$. We can use induction to prove that $\varepsilon_t(\widehat{\mathrm{w}}_t) \leq \epsilon$. $\square$

# D Additional Experimental Results

## D.1 Regression

We consider three regression tasks $T_1, T_2, T_3$, where $T_3$ is the target task. The data of the tasks are vectors in $\mathbb{R}^d$ with $d = 1000$. Set $\mu_1 = (1, 0, \ldots, 0)^\top$, $\mu_2 = (1.5, 0, \ldots, 0)^\top$, $\mu_3 = (2, 0, \ldots, 0)^\top$. The underlying distributions of the three tasks are $\mathcal{D}_1 = \mathcal{N}(\mu_1, I_d), \mathcal{D}_2 = \mathcal{N}(\mu_2, 3I_d), \mathcal{D}_3 = \mathcal{N}(\mu_3, 10I_d)$. For any example z and weight vector w, the squared loss is defined as $\ell(w, z) = \|w - z\|^2$. Since we are using a constant vector w to predict every input, the three formulated tasks are equivalent to three mean estimation problems. We solve the three tasks using regularized ERM. Set $\widehat{w}_0 = 0$. For each task, we incorporate an $\ell_2$ regularization term into the empirical risks, $\lambda \|w_t - \widehat{w}_{t-1}\|^2$, where $\widehat{w}_{t-1}$ represents the optimal weights learned from the previous task $t-1$. For the mean estimation of $\mathcal{N}(\mu, \sigma^2 I_d)$, the regularization parameter is set as $\lambda = \frac{d\sigma^2}{n\|\mu - \widehat{w}_{t-1}\|^2}$ directly without using validation, where $n$ is the sample size of the current task. Here $\lambda$ is set to minimize the test loss in expectation. We fix the sample size $n_1 = n_2 = 1.5K$ for the first two tasks. We choose different sample size of the target task $n_3$ and demonstrate the statistical benefit of our curriculum. We compare six different training methods: learning $T_3$ directly using ERM; learning $T_3$ directly using RERM; learning $T_1, T_3$ sequentially; learning $T_2, T_3$ sequentially; learning $T_2, T_1, T_3$ sequentially; and learning $T_1, T_2, T_3$ sequentially. Table 2 and Figure 3 report the averaged test loss $\|\widehat{w} - \mu_3\|^2$ for all training methods over 5M repetitive runs. Our results show that learning an easier task before solving the target task $T_3$ leads to a smaller expected risk compared with solving $T_3$ directly. The curriculum that learns $T_1, T_2, T_3$ sequentially achieves the smallest expected risk among all methods.

| $n_3$ | $T_3$(ERM) | $T_3$(RERM) | $T_1+T_3$ | $T_2+T_3$ | $T_2+T_1+T_3$ | $T_1+T_2+T_3$ |
|---|---|---|---|---|---|---|
| 1K | 10.000 | 2.857 | 1.803 | 1.677 | 1.329 | 1.326 |
| 2K | 5.000 | 2.222 | 1.528 | 1.436 | 1.173 | 1.171 |
| 3K | 3.333 | 1.818 | 1.325 | 1.255 | 1.050 | 1.048 |
| 4K | 2.500 | 1.538 | 1.170 | 1.115 | 0.950 | 0.948 |
| 5K | 2.000 | 1.333 | 1.047 | 1.003 | 0.868 | 0.866 |
| 6K | 1.667 | 1.176 | 0.948 | 0.912 | 0.798 | 0.797 |
| 7K | 1.429 | 1.053 | 0.866 | 0.836 | 0.739 | 0.738 |
| 8K | 1.250 | 0.952 | 0.797 | 0.771 | 0.688 | 0.688 |
| 9K | 1.111 | 0.870 | 0.738 | 0.716 | 0.644 | 0.643 |
| 10K | 1.000 | 0.800 | 0.687 | 0.668 | 0.605 | 0.604 |
| 11K | 0.909 | 0.741 | 0.643 | 0.626 | 0.571 | 0.570 |
| 12K | 0.833 | 0.690 | 0.604 | 0.589 | 0.540 | 0.539 |
| 13K | 0.769 | 0.645 | 0.570 | 0.557 | 0.512 | 0.512 |
| 14K | 0.714 | 0.606 | 0.539 | 0.527 | 0.487 | 0.487 |
| 15K | 0.667 | 0.571 | 0.512 | 0.501 | 0.465 | 0.464 |
| 16K | 0.625 | 0.541 | 0.487 | 0.477 | 0.444 | 0.444 |
| 17K | 0.588 | 0.513 | 0.464 | 0.455 | 0.425 | 0.425 |
| 18K | 0.556 | 0.488 | 0.444 | 0.435 | 0.408 | 0.407 |
| 19K | 0.526 | 0.465 | 0.425 | 0.417 | 0.392 | 0.391 |
| 20K | 0.500 | 0.444 | 0.407 | 0.401 | 0.377 | 0.377 |

Table 2: Test loss of different training methods under different sample size $n_3$.

## D.2 More Details on Synthetic Datasets Experiment

Here we provide more contents regarding the synthetic dataset experiment as described in Section 6. We aim to investigate whether leveraging curriculum learning can improve the performance of large-margin classifiers when dealing with separable data. The motivation is rooted in the intuition that if we can distinguish between data points that are easy versus hard to classify, we may benefit from a curriculum learning strategy. Specifically, by first training on easy-to-classify points to learn an initial model and then fine-tuning using harder examples, we hypothesize that the model can better generalize to challenging tasks.

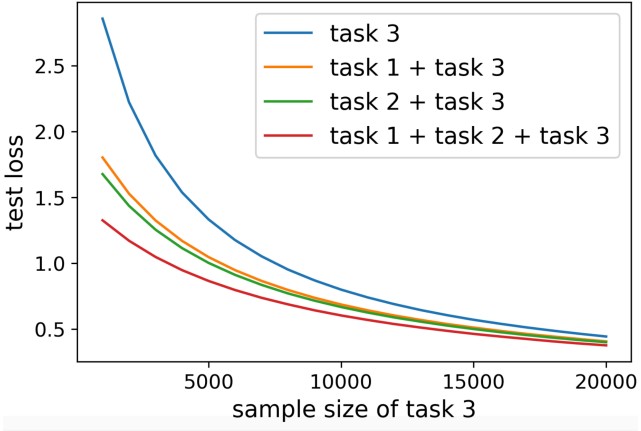

Figure 3: Test loss as a function of the sample size $n_3$.

To test this hypothesis, we consider a binary classification task where data is drawn from a mixture of two distributions, $\mathcal{D}_1$ and $\mathcal{D}_2$. Each distribution is defined as a two-centered Gaussian in $\mathbb{R}^d$ with dimension $d = 100$. For each distribution $\mathcal{D}_i$, $i \in \{1, 2\}$, the two centers are located at the origin and at $[\gamma, 0, \dots, 0]^T$, with the spread determined by the Gaussian noise standard deviation $\sigma$. We generate 1K training samples from each distribution. Distribution $\mathcal{D}_1$, with $\gamma = 3$ and $\sigma = 0.5$, is considered "easy" due to its large margin and low variance. In contrast, $\mathcal{D}_2$ is constructed as a "hard" distribution, with parameter $\gamma \in [0.1, 0.5, 1.0, 2.0]$ and $\sigma \in [0.5, 1.0, 1.5, 2.0]$. Additionally, we generate 400 validation samples and 400 test samples from $\mathcal{D}_2$.

We train the linear model using both logisitic loss and hinge loss, and optimize it with gradient descent for 2K epochs, using a learning rate selected from $\{0.001, 0.01, 0.05, 0.1, 0.5, 1.0\}$. No regularization is applied when training the easy distribution $\mathcal{D}_1$. When fine-tuning on training dataset from $\mathcal{D}_2$, we incorporate an $\ell_2$ regularization term into the loss functions, $\lambda\|\mathbf{w}_2 - \widehat{\mathbf{w}}_1\|^2$, where $\widehat{\mathbf{w}}_1$ is the optimal model weights from previous training and the regularization parameter $\lambda$ is selected from the set $\{10^{-5}, 10^{-4}, 10^{-3}, 10^{-2}, 10^{-1}, 1, 10\}$. Experiments are repeated 10 times, and we report the mean test accuracy along with standard deviation in Figure 4. Our results show that curriculum learning consistently outperforms the baseline, indicating that starting with an easier task facilitates learning on the more challenging target task. Furthermore, the harder the target task (characterized by a smaller margin and larger standard deviation), the more significant the improvement achieved through curriculum learning.

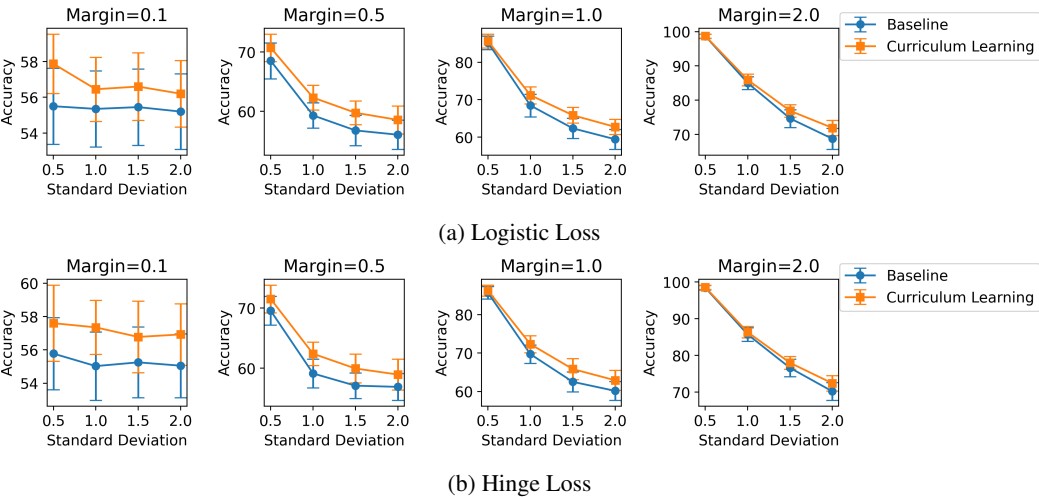

Figure 4: Test accuracy as a function of standard deviation for different margin $\gamma$.

## D.3    More Details on Adversarial Training Experiments

We provide additional details and results on evaluating curriculum adversarial training with $\ell_2$ regularization on MNIST dataset. We consider adversarial examples generated using both $\ell_\infty$-norm perturbation (with budgets $\alpha \in \{0.1, 0.2, 0.3, 0.4\}$) and $\ell_2$-norm perturbation (with budgets $\alpha \in \{1.0, 2.0, 3.0, 4.0\}$). All adversarial examples are generated using 10-step PGD with a step size of $\alpha/5$. We hold out 20% of the training data as a validation set and use 10-step PGD adversarial examples, crafted with the same perturbation budget $\alpha$, for hyper-parameter tuning and model selection. No regularization is used for $t = 1$. From $t \geq 2$, we incorporate $\ell_2$ regularization of the form $\lambda \|w_t - \widehat{w}_{t-1}\|^2$, where $\widehat{w}_{t-1}$ is the previous model and $\lambda \in \{10^{-5}, 10^{-4}, 10^{-3}, 10^{-2}\}$. We use previous model $\widehat{w}_{t-1}$ as initialization for $t$. The setting has been described in Section 6. We report both standard and robust test accuracy under PGD attacks of size $\alpha$ over three repetitive runs in Table 3 for $\ell_\infty$-attacks and Table 4 for $\ell_2$-attacks. We observe that curriculum adversarial training maintains performance for small $\alpha$ and provides improvements for larger $\alpha$ ($\alpha \geq 0.3$ for $\ell_\infty$-attacks and $\alpha \geq 3.0$ for $\ell_2$ attacks).

| T $\diagdown$ $\alpha$ | 1 | | 2 | | 3 | |
|---|---|---|---|---|---|---|
| | nat acc | pgd acc | nat acc | pgd acc | nat acc | pgd acc |
| 0.1 | 99.18±0.07 | 96.07±0.02 | 99.27±0.07 | 95.65±0.18 | 99.36±0.03 | 95.74±0.14 |
| 0.2 | 98.80±0.03 | 94.73±0.22 | 98.86±0.15 | 94.60±0.93 | 98.67±0.05 | 94.38±0.23 |
| 0.3 | 98.27±0.46 | 92.77±1.20 | 98.77±0.15 | 94.74±0.12 | 98.23±0.15 | 93.61±0.87 |
| 0.4 | 11.35±0.00 | 11.35±0.00 | 98.39±0.29 | 95.54±0.41 | 98.52±0.14 | 95.63±0.12 |

Table 3: Standard (nat acc) / robust (pgd acc) accuracy under $\ell_\infty$ PGD attack of size $\alpha$ (MNIST).

| T $\diagdown$ $\alpha$ | 1 | | 2 | | 3 | |
|---|---|---|---|---|---|---|
| | nat acc | pgd acc | nat acc | pgd acc | nat acc | pgd acc |
| 1.0 | 99.32±0.05 | 94.53±0.16 | 99.26±0.05 | 94.44±0.03 | 99.37±0.04 | 93.99±0.39 |
| 2.0 | 98.38±0.13 | 76.23±0.39 | 98.51±0.10 | 76.04±0.30 | 98.40±0.08 | 76.56±0.27 |
| 3.0 | 94.35±0.70 | 52.11±0.60 | 94.87±0.30 | 52.53±0.35 | 94.06±1.29 | 52.65±0.86 |
| 4.0 | 89.12±2.79 | 31.47±0.41 | 87.62±1.53 | 31.93±0.94 | 84.55±1.35 | 32.00±0.80 |

Table 4: Standard (nat acc) / robust (pgd acc) accuracy under $\ell_2$ PGD attack of size $\alpha$ (MNIST).

## D.4    Noisy MNIST

We construct a noisy MNIST dataset by adding Gaussian noise to each example, sampled from the distribution $\mathcal{N}(0, \sigma^2 I_{784})$. Our goal is to find a model that perform well on the noisy MNIST test data. To perform curriculum learning, we manually categorize the digits into four groups: [1,4,7], [3,8,0], [6,9], [2,5] and create four tasks as follows: 1). train on digits [1,4,7]; 2). train on digits [1,4,7]∪[3,8,0]; 3). train on digits [1,4,7]∪[3,8,0]∪[6,9]; 4). train on all digits. The reason for selecting these categories is based on the visual similarity in the shape of the digits.

We consider three different architectures: a linear model, a two-layer ReLU network with a hidden width of 100, and a convolutional neural network (CNN). The CNN consists of two convolutional layers followed by max-pooling and two fully connected layers with ReLU activations. The first and second convolutional layers have [input channel, output channel, kernel size] = [1, 10, 5] and [10, 20, 5], respectively. The first and second fully connected layers have dimensions [320, 100] and [100, 10], respectively. For each task, we train the model using cross-entropy loss and optimize it with stochastic gradient descent (SGD) for 200 epochs, using a batch size of 128 and a learning rate selected from $\{10^{-4}, 10^{-3}, 10^{-2}, 10^{-1}\}$, with a weight decay of $10^{-4}$. No regularization is applied during the first task. From the second task onward, we incorporate an $\ell_2$ regularization term into the loss functions, $\lambda \|w_t - \widehat{w}_{t-1}\|^2$ for task $t \geq 2$, where $\widehat{w}_{t-1}$ represents the optimal model weights from the previous task $t - 1$. The regularization parameter $\lambda$ is selected from $\{10^{-5}, 10^{-4}, 10^{-3}, 10^{-2}, 10^{-1}, 1\}$. We randomly set 20% of the training data aside as the validation set and use the validation accuracy to select the optimal hyperparameters for each task and to determine the best-performing model checkpoint. The optimal model weights from task $t - 1$ are used both to initialize model training and as the reference point for the $\ell_2$ regularizer in task $t$. Table 5 reports the averaged results over three runs, including standard deviations. Our results demonstrate that curriculum learning consistently outperforms the baseline, particularly when the

noise level $\sigma$ is higher. This indicates that curriculum learning is especially beneficial in more challenging settings where the data is noisier.

| Model | $\sigma$ | Baseline | Curriculum |
|---|---|---|---|
| Linear | 0.0 | 92.04±0.05 | 92.34±0.10 |
| | 1.0 | 68.93±0.63 | 70.69±0.38 |
| | 2.0 | 42.18±0.65 | 44.63±0.29 |
| Two-layer ReLU Network | 0.0 | 97.87±0.14 | 97.85±0.05 |
| | 1.0 | 80.55±0.24 | 81.26±0.13 |
| | 2.0 | 45.79±0.91 | 46.43±0.26 |
| Convoluted Network | 0.0 | 99.00±0.06 | 99.04±0.04 |
| | 1.0 | 84.48±0.29 | 85.08±0.11 |
| | 2.0 | 46.66±0.65 | 48.33±0.15 |

Table 5: Accuracy on the $\sigma$-noisy test data under different $\sigma$ and different model architectures.

