# OpenReview forum: "When Does Curriculum Learning Help? A Theoretical Perspective"
_NeurIPS.cc/2025/Conference — NeurIPS 2025 poster_

### Official Review · Reviewer_kjZ2 · 2025-06-22

**Clarity:** 3
**Significance:** 3
**Originality:** 3
**Rating:** 4
**Confidence:** 4

**Summary:**

The paper presents several key contributions to the theoretical understanding of curriculum learning—the strategy of training a model on progressively harder tasks. The authors introduce a formal framework for analyzing multi-task curriculum learning. This framework is centered on Biased Regularized Empirical Risk Minimization (RERM) by using an easy task to provide better inductive biases  to regularize the learning process for the next, more difficult task.

Some theoretical contirbutions of this paper:
The (r, α) Condition which connects task similarity and empirical risk. This condition mathematically formalizes the similarity between consecutive tasks. It asserts that if a model performs well on one task, a slightly perturbed version of that model will also perform reasonably well on the subsequent task, so as long as the r is small enough.

The analysis provides rigorous proofs and performance guarantees for different types of optimization problems.
For Convex Problems:
The paper provides proofs and excess risk bounds showing that a good curriculum (one with a small r) can significantly reduce the sample complexity required for subsequent tasks.
These guarantees are proven for both the RERM framework and the more practical Stochastic Gradient Descent (SGD) method.
For Non-Convex Problems:
The analysis is extended to non-convex settings, which are highly relevant for modern deep learning.
The authors propose an ERM-based algorithm and establish generalization guarantees, demonstrating that a well-designed curriculum improves learning efficiency even in these complex scenarios.

**Questions:**

1. [Eq 2] how do you define this norm when there are arch changes, tasks need not have errors at the same scale
2. In your work you mentioned about the scale of loss being different in different subsequent tasks, does that not affect the $\alpha$ parameter? how if at all any rescaling operation done? if not then despite being a optimal curriclum, the scale difference might indicate a high α
3. Can you provide some experimental evidence for curricula where the task does not necessarily share the same arch. or target label set, basically increase the complexity of the underlying set of tasks in  the curriculum to show if it is possible to empirically validate the theoretical findings to a more realistic setting.

**Ethical Concerns:**

["NO or VERY MINOR ethics concerns only"]

**Final Justification:**

The authors have successfully adressed the majority of my concerns as mentioned in the rebuttal.

**Limitations:**

This work specifies the generalization on the subsequent task given a particular curriculum. The harder and perhaps more important question is what is the optimal curriculum. given the inability to define the (r, α) criterion for a general setting, the applicability of this method seems limited.

How can we identify the (r, α) criteria for a given curriculum in a general setting?

It's unclear how these results and the underlying theoretical dynamics would scale to much larger and more complex real-world scenarios. For instance, training massive models like large language models (LLMs) or state-of-the-art vision models on datasets like ImageNet involves far more complex dynamics. While the principles might hold, the practical benefits could be different or harder to achieve, and the cost of exploring different curricula would be prohibitively high.


The process of ordering tasks remains largely a heuristic, domain-specific art that relies on human intuition. For example, in their experiments, the authors rely on common-sense curricula: increasing the difficulty of a classification task by reducing the margin, or increasing the strength of an adversarial attack. The framework doesn't offer a general algorithm for discovering this optimal ordering automatically.

**Paper Formatting Concerns:**

1. Fig 1, task index on the minima not mentioned.

**Quality:**

3

**Strengths And Weaknesses:**

Strengths of this paper:

1. This paper, to my knowledge is one of the first to present the notion of (r,α) condition of task similarity to empirical risk.
2. Authors provides guarantees for both convex and non-convex optimization problems (thm 4.1, 4.2). Since most modern deep learning involves highly non-convex landscapes, this makes the paper's conclusions far more relevant and impactful for practitioners working with neural networks.
3. The authors extend their analysis from the more abstract Biased RERM to Stochastic Gradient Descent (SGD), which is the optimizer used in the vast majority of real-world machine learning.

Weakness:
This work, to my understanding hinges on the (r, α) condition. which in this paper works only when the task sequence have the same domain and target space and the arch. is same. For a more general setting this might not be possible

The experimentation is a bit weak, for a general setting of sequential how can we identify the rho and alpha, While the experiments are well-designed to support the theoretical claims, they are conducted on relatively small-scale and controlled environments (synthetic data and MNIST).

---

> ### Author Rebuttal · Authors · 2025-07-29
>
> **(W1) Applicability Beyond Shared Architecture/Label Space:**
>
>  We would like to clarify that while Eq(2) is initially stated assuming a shared parameter space across tasks, our framework can accommodate architectural changes.
>
>
> As discussed (lines 121–131), we allow for the existence of a mapping $\phi_t : \mathcal{H}_{t-1} \to \mathcal{H}_t$ that transfers the solution from task $t-1$ into the parameter space of task $t$. This captures cases where the architecture changes across tasks, for instance, embedding a smaller network into a larger one by initializing new weights carefully (e.g., identity padding, layer-wise expansion). In such settings, we replace $w$ with $\phi_t(w)$ in Equation (2), allowing our $(r, \alpha)$-condition to remain meaningful even when the architecture or hypothesis class evolves.
> We assume $\phi_t$ is the identity for simplicity in our core theorems, but the underlying idea is that a good solution from task $t-1$ provides useful prior knowledge, potentially through transfer mappings, for task $t$. We will make this interpretation clearer in the revision.
>
>
> Furthermore, in Appendix D, we present an experiment on Noisy MNIST where the label space expands over time.
>
>
> **(W2) Weak experimentation, need to know (r, α) in practice, weak experiments**:
>
> While the experimental section is limited in scope, it includes a nontrivial application to adversarial robustness, a realistic and practically relevant setting for both NLP/vision, demonstrating how our theory can guide curriculum design. The additional experiments on synthetic and MNIST data serve as proof-of-concept to validate key ideas in controlled settings. We view these results as complementary to the theory, not as the central contribution.
>
>
> Also, (r, α) are not parameters to be identified or estimated or tuned in practice. Rather, it is a theoretical condition that characterizes a good curriculum, similar in spirit to margin assumptions or noise conditions in classical learning theory. In practice, the only parameter that needs tuning in our biased RERM algorithm is the regularization weight. This can be selected using standard validation techniques, which is exactly what we do in our experiments. We will emphasize this more clearly in the revised version to avoid confusion between theoretical assumptions and tunable hyperparameters.
>
>
> **(Q1a) On the Norm in Equation (2) Under Architecture Changes and Loss Scaling:**
>
> In Eq(2), the norm $\|\|w - w'\|\|$ is defined over the shared or mapped parameter space across consecutive tasks. When the architecture changes, we explicitly address this in our paper (lines 121–131) by introducing a mapping $\phi_t : \mathcal{H}_{t-1} \to \mathcal{H}_t$​ that embeds the solution from task $t-1$ into the space of task $t$. In that case, the comparison in Equation (2) becomes $\|\|\phi_t(w) - w_t^\star\|\|$, and the norm is taken in the ambient space of $\mathcal{H}_t$. This allows for increasing model capacity, e.g., by adding neurons or layers, while preserving the interpretation of prior solutions as inductive biases for future tasks.
>
>
> Regarding loss scaling, the magnitude of $\alpha_t$ is indeed influenced by the relative scale of loss functions across tasks. However, as discussed in our response to point (2), the $(r, \alpha)$-condition is invariant to affine rescaling of the loss. That is, multiplying a loss function $\ell_t$ by a positive constant rescales $\alpha_t$ accordingly, without changing the underlying minimizer or risk structure. We assume comparable loss scales across tasks for simplicity, but this can be adjusted in practice through normalization if needed.
>
>
> In summary, both norm comparability (via $\phi_t$) and loss scale (via normalization) are accounted for in our framework. We will clarify these assumptions more explicitly in the revision.
>
> **(Q1b) Scale Mismatch and Loss Normalization in the $(r, \alpha)$-Condition:**
>
> We assume for simplicity that the $\alpha_t$ in Equation (2) satisfies $\alpha_t < 1$, and that loss functions across tasks are scaled similarly. This assumption is not essential and can be relaxed by a straightforward rescaling of the losses.
>
>
> As we explain (lines 142–147), if the loss for task $t$ differs in scale from previous tasks, this can be absorbed by appropriately scaling the loss function $\ell_t$ and adjusting $\alpha_t$ accordingly. For example, if the loss at task $t$ is twice that of task $t+1$ (say, due to arbitrary units in squared loss), then multiplying $\ell_t$ by a constant would reduce $\alpha_t$ without affecting the minimizer or theoretical guarantees. The key insight remains: a small value of $r_t$, indicating a good initialization,  is far more important than the absolute value of $\alpha_t$, which reflects loss scaling.
>
>
> In practice, we do not rescale losses across tasks. But for theoretical clarity, assuming a constant $\alpha < 1$ allows us to streamline the presentation of our results without loss of generality. If needed, our theorems could be extended to carry task-dependent $\alpha_t$ values throughout. We will clarify this in the revised version.
>
>
> **(Q2) Experiments on Complex or Heterogeneous Tasks:**
>
> While our primary focus is theoretical, we do include some experiments that begin to address this setting. In Appendix D, we present an experiment on Noisy MNIST where the label space expands over time.
>
>
> We cite (lines 31–32) several prior empirical works study curriculum-style training where model capacity is increased over time. These can be interpreted as tasks with varying architectures, where parameters from the previous model are carefully mapped into a larger one, consistent with our theoretical allowance for task mappings (via $\phi_t$).
>
>
>
>
> **(L1) On Identifying the $(r, \alpha)$ Condition in Practice:**
>
>  We agree that directly computing or verifying the $(r, \alpha)$-condition in general settings may be difficult. However, the purpose of our framework is not to prescribe explicit values of $r$ or $\alpha$, but rather to provide a theoretical lens that characterizes when curriculum learning is effective. Just as assumptions like margin conditions or realizability are used to structure classical learning theory, the $(r, \alpha)$-condition helps formalize what it means for one task to serve as a good precursor to another.
>
>
> While we do not expect practitioners to estimate $(r, \alpha)$ precisely, the condition informs practical design principles. Specifically:
>
> - Good curricula should transition gradually between tasks, ensuring that task $t+1$ is not too far (in risk or parameter space) from task $t$.
>
>
> - This can be achieved, for instance, by ordering training examples by increasing difficulty, or incrementally expanding the hypothesis class, strategies reflected in our adversarial robustness experiment and discussed in Section 4.4 (lines 269–284).
>
> In short, the $(r, \alpha)$-condition is best viewed as a guiding structural principle for designing curricula, not a parameterization to be optimized. We will clarify this role more explicitly in the revised version.
>
> **(L2) Scope and Complexity of Experimental Evaluation:**
>
> This is primarily a theoretical paper, aimed at providing a rigorous foundation for curriculum learning, a widely used but under-theorized paradigm. We introduce a novel sufficient condition for curriculum learning and establish generalization guarantees for both convex and non-convex settings, a significant step forward in understanding curriculum structure.
>
>
> While our experimental validation is limited in scope, it serves to complement and illustrate the theoretical results in controlled settings:
> - Our adversarial robustness experiment demonstrates how the theory can guide curriculum design in a practically relevant deep learning scenario. This setup is applicable across both vision and NLP domains where adversarial training is widely studied.
>
>
> - In Appendix D, we present a Noisy MNIST experiment that progressively increases the label space,  offering a simple empirical example where tasks differ structurally (i.e., increasing classification complexity), yet the curriculum remains beneficial.
>
> We acknowledge that extending our framework to more diverse curricula, including tasks with shifting data distributions, output spaces, or architectures, is important future work.
>
> **(L3) On the Role of Heuristics in Task Ordering and the Absence of an Algorithm for Optimal Curricula:**
>
> We appreciate the reviewer’s observation and agree that designing effective task orderings often involves heuristics or domain knowledge. However, this is not a limitation unique to our work, but rather a fundamental feature of curriculum learning itself. As the reviewer notes, curricula in our experiments (e.g., increasing adversarial strength or classification difficulty) are designed using common sense. This “common sense” corresponds precisely to inductive bias or prior knowledge, which is essential in any learning setting. Our contribution is to formalize this inductive bias through the $(r, \alpha)$-condition.
>
> There is no free lunch in machine learning: learning always requires assumptions. In curriculum learning, the key assumption is that the early tasks help prepare the learner for the final task. Our framework does not claim to discover an optimal curriculum automatically; rather, it aims to formalize what makes a curriculum effective, and to provide a theoretical foundation for analyzing or comparing curricula.
>
> Indeed, the $(r, \alpha)$-condition offers both predictive and diagnostic power: it not only guides curriculum design (e.g., gradual expansion of the hypothesis class or task difficulty), but also helps explain why a curriculum may fail, e.g., if early tasks do not sufficiently prepare the learner, generalization on later tasks will suffer. This is a fundamentally different perspective from multitask learning, which assumes a fixed set of tasks to be learned jointly or reordered.

---

> > ### Comment · Reviewer_kjZ2 · 2025-08-05
> > **Response to rebuttal**
> >
> > I have read the rebuttal by the authors and my concerns are more or less addressed by it.
> >
> > My concerns W1, Q1, L1, L2, L3 have been fully addressed to my satisfaction.
> > I still feel the rigour of experimentation required to address Q2 and W2 might be a bit hard but I understand the practical limitations yet. and based on this I am willing to increase my score.

---

> > > ### Author Response · Authors · 2025-08-07
> > > **Thank you!**
> > >
> > > We would like to sincerely thank you for your thoughtful review, your follow-up engagement, and for considering our rebuttal in detail. We are pleased that our clarifications addressed most of your concerns (W1, Q1, L1, L2, L3) to your satisfaction, and we appreciate your understanding of the practical limitations around Q2 and W2.
> > >
> > >
> > > As you finalize your assessment, we kindly ask that you take our responses into account in full. If you have any remaining questions, comments, or suggestions, whether on technical aspects, presentation, or possible extensions, we would be happy to address them. Your feedback has been valuable in strengthening the clarity and impact of our work, and we greatly appreciate your constructive engagement throughout the process.
> > >
> > >
> > > Thank you again for your time and consideration.

---

### Official Review · Reviewer_DGsF · 2025-06-28

**Clarity:** 4
**Significance:** 3
**Originality:** 4
**Rating:** 6
**Confidence:** 4

**Summary:**

The paper introduces (for the first time?) a theoretical analysis of curriculum learning. It studies a scenario where the learner goes through a sequence of tasks where the parameters found at the end of training for each task is a good initialization for the next task (and not far from a good solution for the next task). This is a scenario that is reminiscent of continuation methods in optimization, but the paper provides a number of theoretical bounds on generalization at each step, given the expected error at the previous step. Several other theoretical assumptions are leveraged to obtain a diversity of results, e.g., if the losses are convex in the parameters, if it is Lipschitz, and if it is smooth. Some of the results apply in the SGD non-convex case typical of neural networks and the paper describes numerical simulations which support the theoretical results.

**Questions:**

Is there absolutely no previous work on a learning-theoretical analysis of curriculum learning?

How are the proof techniques related to previous work in learning theory?

**Ethical Concerns:**

["NO or VERY MINOR ethics concerns only"]

**Final Justification:**

The authors have satisfactorily answered my questions, so I maintain my rating.

**Limitations:**

In the conclusion, briefly.

**Paper Formatting Concerns:**

none I could spot.

**Quality:**

4

**Strengths And Weaknesses:**

ular pre-training followed by successive fine-tuning stages can be seen as instances of curriculum learning that plausibly fit some of the scenarios studied in the paper.

As far as I know, this is the first learning-theoretical analysis of curriculum learning, but I would have liked to see confirmation or not in the paper, along with a review of relevant related work, if only in terms of the theoretical tools being used, as well as non-theoretical work on curriculum learning.

---

> ### Author Rebuttal · Authors · 2025-07-29
>
> **(Q1) Is There Absolutely No Previous Learning-Theoretic Work on Curriculum Learning?**
>
> We appreciate the reviewer raising this important question. While our work is, to the best of our knowledge, the first to offer a general learning-theoretic framework for curriculum learning with generalization guarantees across both convex and non-convex regimes, we do cite and build upon several earlier theoretical efforts:
> - **Pentina et al. (2015)** (Curriculum Learning of Multiple Tasks) consider a similar sequential task setup and propose a biased regularizer in an SVM setting. Their analysis focuses on minimizing the average population risk over all tasks, whereas our objective is to minimize the population risk on a final target task. Our framework is also more general in that it applies to arbitrary convex losses and allows for multiple tasks with general data distributions.
>
>
> - **Weinshall et al.** develop curriculum strategies based on difficulty scores and show improved convergence rates. Their analysis is rooted in optimization and focuses on acceleration rather than generalization. Our focus is on sample complexity and excess risk.
>
>
> - **Xu and Tewari (2022)** and **Cohen et al. (2022)** analyze simplified two-task curricula in mean estimation problems to highlight statistical benefits. Our results extend this line of work to general hypothesis classes and arbitrary sequences of tasks.
>
>
> - **Abbe et al. (2023)** and **Sarao Mannelli et al. (2020)** both study very specific structured tasks (parity functions, teacher-student models with Gaussian inputs). While they propose concrete two-phase curricula and analyze special cases (often using statistical physics), our framework allows for arbitrary-length curricula with minimal assumptions on the data distribution, hypothesis class, or task structure.
>
>
> In summary, while some elements of our approach,  such as the use of biased regularization, have appeared in specific contexts, our contribution is the first to unify and generalize these ideas into a broadly applicable learning-theoretic foundation for curriculum learning. We will incorporate this discussion more explicitly in the related work section.
>
>
> **(Q2) Connection to Proof Techniques in Learning Theory:**
>
> Most of our proof techniques are grounded in standard tools from statistical learning theory. In particular, we use algorithmic stability to analyze biased RERM (especially in the SGD setting), and uniform convergence arguments for ERM-based excess risk bounds.
>
> The key novelty lies in how we integrate these classical techniques with the (r,α)-condition, a structural assumption that captures task-to-task compatibility in a curriculum. This allows us to propagate guarantees across tasks and formally characterize when curriculum learning improves generalization. We will clarify these connections and include the relevant citations in the revision.

---

> > ### Author Response · Authors · 2025-08-07
> > **Thank you!**
> >
> > We would like to express our sincere thanks for your thoughtful and encouraging review of our work. Your careful reading, clear summary, and recognition of both the novelty and potential impact of our framework mean a great deal to us.
> >
> >
> > Your positive assessment reinforces our confidence that these ideas can serve as a foundation for further progress in the area. If you have any additional thoughts or suggestions, whether on strengthening the presentation, highlighting certain aspects more clearly, or potential future directions, we would be glad to hear them.
> >
> >
> > Thank you again for your generous and insightful feedback.

---

### Official Review · Reviewer_CsMF · 2025-07-01

**Clarity:** 3
**Significance:** 2
**Originality:** 2
**Rating:** 4
**Confidence:** 4

**Summary:**

In the present paper, the authors develop a biased-regularised ERM view of curriculum learning. Given a sequence of $T$ supervised tasks, each task $t$ is learned by minimising the empirical loss plus an $L_2$ penalty that keeps the parameters close to the solution of task $t-1$. The analysis is based on the definition of an $(r, \alpha)$ “good curriculum” condition, used for proving excess-risk bounds for convex, Lipschitz, and smooth losses. The results are then extended, under some assumptions, to SGD and a non-convex setting. The empirical evaluation is based on experiments on synthetic data and MNIST (including adversarial training).

**Questions:**

On the tightness of the bounds: Could the authors overlay the analytical excess-risk curves on the experimental plots, so readers can see how loose/tight they are?

On choosing $(r, \alpha)$ in practice: Can the authors comment on how one would estimate or tune $r$ when the optimal risks or parameter distances are unknown?

On the relation to earlier work: Can the authors discuss the connections to Sarao Mannelli et al. (2020) and clarify the novelty of their practical findings?

On the connection to continual-learning: The drift-per-task setting resembles CL more than classic curriculum. How does this method fare in settings where catastrophic forgetting can take place?

**Ethical Concerns:**

["NO or VERY MINOR ethics concerns only"]

**Final Justification:**

Overall, assuming the related literature is incorporated and that the authors will clarify the distinction between theoretical assumptions and practical implementation, potentially highlighting the gaps, I am inclined to raise my score to a 4:borderline accept.

**Limitations:**

The discussion of the limitations is very limited (a comment in the Conclusions).

**Quality:**

3

**Strengths And Weaknesses:**

**Strengths**
- The work is technically solid: assumptions are carefully stated, proofs are complete, and empirical evaluation seems sufficient (especially considering the long appendix).
- The idea of introducing the $(r, \alpha)$ compatibility metric seems to allow a simple formalization of task similarity.
- Overall, the paper is clearly written.

**Weaknesses**
- Some relevant references to theoretical works on curriculum learning are missing (e.g., the recent line by Abbe et al). In particular, a similar curriculum protocol with elastic penalties was analysed analytically and experimentally in "An Analytic Theory of Curriculum Learning" (Sarao Mannelli et al., 2020), which the paper does not cite.
- The derived generalisation bounds depend on (potentially unknown) global Lipschitz, smoothness, and local-geometry constants. This makes it hard to judge their tightness or to understand how they can be used in practice (the bound curves are not plotted alongside the learning curves in the empirical experiments).
- It is a bit unclear whether the analyzed setting is truly connected to curriculum learning or rather represents a generic continual/transfer learning setting.
- The $(r, \alpha)$ condition and the regularisation schedule presume task knowledge that a learner rarely has access to.

---

> ### Author Rebuttal · Authors · 2025-07-29
>
> **(W1) Missing References (Sarao Mannelli et al., Abbe et al.):**
>
>  We thank the reviewer for highlighting these works. We will cite and discuss both in the revised version.
>
> The paper by Abbe et al. (Provable Advantage of Curriculum Learning on Parity Targets with Mixed Inputs) studies a highly structured learning task and proves that a two-phase curriculum improves generalization in that specific setting. While their setup is quite different from ours, the underlying motivation aligns with our goal of providing rigorous conditions under which curriculum learning is beneficial.
>
> Similarly, An Analytical Theory of Curriculum Learning in Teacher-Student Networks (Sarao Mannelli et al., 2020) analyzes a teacher-student model with inputs drawn from Gaussian distributions and labels generated by a perceptron. They focus on a specific curriculum involving low-variance inputs and introduce elastic penalties between tasks, conceptually related to our regularization framework. However, their analysis relies on tools from statistical physics and is restricted to specialized synthetic settings. In contrast, our work provides generalization guarantees for convex and non-convex losses in a more general supervised learning framework.
>
> We appreciate the connections and will incorporate a comparative discussion in the related work section.
>
>
> **(W2) Tightness of Bounds and Practical Use of Constants:**
>
>  We would like to clarify that the global Lipschitz and smoothness constants used in our excess risk bounds are not unknown once the hypothesis class and input space are specified. These constants are standard in learning theory (see, e.g., Understanding Machine Learning by Shalev-Shwartz and Ben-David) and are determined by the choice of loss function and input domain.
>
> Moreover, these constants are not needed to implement our algorithm. In practice, the regularization parameters can be tuned using validation, as we do in our experiments.
>
> Our theoretical contribution shows that under the (r, α) condition, curriculum learning leads to improved generalization. The key driver of this improvement is the parameter r: when r is small, our excess risk bounds are small, regardless of whether the constants are numerically tight. The purpose of our analysis is to provide structural insight into when curriculum learning helps, not necessarily to optimize constant factors in the bound. We will clarify this point in the revision.
>
> **(W3) Connection to Curriculum vs. Continual and Transfer Learning:**
>
>
> We appreciate the reviewer’s observation and agree that at a surface level, our regularization-based formulation might resemble continual or transfer learning. However, the underlying goals are fundamentally different:
>
> - In continual learning, the aim is to learn a sequence of tasks without forgetting earlier ones. In contrast, curriculum learning is focused on improving generalization on a single target task through a structured progression of tasks.
> - In transfer learning, prior tasks provide general knowledge for initializing or regularizing learning on a target task, but there is no explicit task sequencing. Curriculum learning, by contrast, uses a deliberate ordering of tasks to facilitate smoother optimization and improved generalization.
>
> Our work models curriculum learning as a form of biased RERM, where earlier tasks shape the inductive bias to improve performance on later tasks, particularly the final one. While some structural similarities exist (e.g., between our regularization and knowledge retention), our framework is tailored to curriculum learning and does not seek to address catastrophic forgetting or full-task retention. We will make this distinction clearer in the revised version.
>
> **(W4) Assumptions on Task Knowledge for Regularization Schedule:**
>
>  We appreciate the reviewer’s concern. The task knowledge referenced in our theoretical analysis, such as optimal risks or parameter distances between tasks, is used solely to derive generalization bounds and to formalize the conditions under which a curriculum is effective. These assumptions are not required in practice to implement the algorithm.
>
> In the biased RERM algorithm, the regularization weights can be tuned using standard techniques such as validation, which is precisely what we do in our experiments. The goal of our analysis is to provide a framework that explains why and when curriculum learning helps, not to prescribe a fixed regularization schedule dependent on quantities that are inaccessible during training.
>
> We also note that assuming Lipschitz continuity and smoothness is standard in the learning theory literature (see, e.g., Understanding Machine Learning by Shalev-Shwartz and Ben-David), and is necessary to make any meaningful theoretical claim about generalization.
>
> We will clarify this distinction between theoretical assumptions and practical implementation in the revision.
>
> **(Q1) On the Tightness of Excess Risk Bounds:**
>
> As is common in learning theory, our upper bounds are distribution-free worst-case guarantees, and thus often significantly looser than empirical performance on specific datasets. This does not diminish their value: they provide a structural explanation of when and why a curriculum helps. We will clarify this distinction and, where feasible (e.g., in synthetic settings), consider illustrating the qualitative alignment between bound behavior and learning curves in the revised version.
>
> **(Q2) On Choosing or Estimating (r, α)  in Practice:**
>
> We appreciate this question and would like to clarify that (r, α) are not parameters to be estimated or tuned in practice. Rather, it is a theoretical condition that characterizes a good curriculum, similar in spirit to margin assumptions or noise conditions in classical learning theory. We assume the (r, α) condition in order to derive generalization bounds and to understand why certain curricula are effective.
>
> In practice, the only parameter that needs tuning in our biased RERM algorithm is the regularization weight. This can be selected using standard validation techniques, which is exactly what we do in our experiments. We will emphasize this more clearly in the revised version to avoid confusion between theoretical assumptions and tunable hyperparameters.
>
> **(Q3) On Relation to Sarao Mannelli et al. (2020):**
>
>  We thank the reviewer for raising this connection. We will cite “An Analytical Theory of Curriculum Learning in Teacher-Student Networks” (Sarao Mannelli et al., 2020) and clarify the distinctions.
>
> That work analyzes a highly specific teacher-student setup where labels are generated by a one-layer perceptron and inputs are drawn from Gaussian distributions. The curriculum is constructed using low-variance data to form an auxiliary task, and elastic penalties are introduced primarily to preserve long-term memory, a setting more aligned with continual learning.
>
> In contrast, our paper develops a general theoretical framework for curriculum learning based on biased regularized ERM. We make minimal assumptions (e.g., Lipschitz/smooth losses) and provide generalization guarantees via the (r,α) condition. While their use of elastic penalties bears some similarity to our regularization approach, their analysis relies on statistical physics techniques and is primarily empirical.
>
> Furthermore, our notion of auxiliary task design (e.g., Sections 4.3–4.4) is more general: we define task difficulty via local Lipschitz properties or population risk, and we discuss practical approaches for identifying good subsets (e.g., using pretrained networks). We believe this broader formulation and its theoretical grounding represent a novel contribution.
>
> **(Q4) On the Connection to Continual Learning and Catastrophic Forgetting:**
>
> We appreciate the question and would like to clarify this distinction. In curriculum learning, as modeled in our paper, the objective is to learn a single target task more effectively by leveraging a sequence of progressively structured tasks. At each stage, the learner focuses on the current task, potentially using prior solutions to guide learning, but not with the goal of retaining performance on earlier tasks. As such, catastrophic forgetting can occur, and is not viewed as a failure mode in this context.
>
> In contrast, continual learning explicitly aims to preserve knowledge across tasks and avoid forgetting. Our formulation does not attempt to maintain performance on previous tasks, and instead embraces the idea that the tasks may drift significantly, with the final target task being quite different from the initial ones. The prior knowledge is used only as an inductive bias to facilitate learning the current task, not as a memory mechanism.
>
> We will clarify this distinction more explicitly in the revised version.

---

> > ### Comment · Reviewer_CsMF · 2025-08-05
> > **Comment**
> >
> > I would like to thank the reviewers for their thorough answers. I have a follow-up question:
> >
> > > In practice, the only parameter that needs tuning in our biased RERM algorithm is the regularization weight. This can be selected using standard validation techniques, which is exactly what we do in our experiments. We will emphasize this more clearly in the revised version to avoid confusion between theoretical assumptions and tunable hyperparameters.
> >
> > In many realistic scenarios, where the training data is given but without an explicit difficulty level, a curriculum strategy requires the imputation of the difficulties associated with the available samples, and then with the organization in training phases of the learning process. It would seem important if your theoretical framework could be used to inform these two steps, somehow exploiting the $(r,\alpha)$ conditions to choose how to order the samples and the splits. Do I understand from your answer that these sub-tasks are assumed to be dealt with *a priori*, and cannot be guided by your theory?
> >
> > ---
> >
> > Overall, assuming the related literature is incorporated and that the authors will clarify the distinction between theoretical assumptions and practical implementation, potentially highlighting the gaps, I am inclined to raise my score to a 4:borderline accept.

---

> > > ### Author Response · Authors · 2025-08-05
> > > **Response to the follow-up comment from Reviewer CsMF**
> > >
> > > We agree with the reviewer that in realistic settings, the training data is typically available without an explicit notion of difficulty. However, our theoretical framework does offer guidance for constructing curricula in such cases. A simple heuristic inspired by our results is as follows:
> > >
> > >
> > > - (a) Begin with a simple hypothesis class, for instance, linear predictors over a fixed feature map with a small norm constraint, and call this $\mathcal{H}_1$​. Train a model from this class and score each training example by the model’s confidence (e.g., positive margin for correct predictions, negative for incorrect ones).
> > >
> > >
> > >
> > >
> > > - (b) Use this scoring to define an initial dataset $D_1$ consisting of examples that are correctly predicted with high confidence. These form the “easy” examples under $\mathcal{H}_1$​.
> > >
> > >
> > > - (c) Then iteratively expand $D_1$ by adding examples that are slightly harder to learn, and simultaneously enlarge the hypothesis class (e.g., by relaxing the norm constraint) to $\mathcal{H}_2,  \mathcal{H}_3,$ etc. The size of each expansion depends on the task/data, but also the user's choice of $(r, \alpha)$.
> > >
> > >
> > > This iterative procedure yields a curriculum $\\{ (D_t, \mathcal{H_t}) \\}_{t=1}^T$, where each phase reflects an incremental increase in task difficulty and model capacity. Our framework not only motivates this strategy but also provides conditions under which it is theoretically sound.
> > >
> > > These ideas are discussed in lines 269–284 of the paper and are operationalized in our adversarial robustness experiment, where the curriculum reflects increasing attack strength.

---

> > > > ### Author Response · Authors · 2025-08-07
> > > > **Thank you!**
> > > >
> > > > We would like to thank the reviewer for the detailed feedback and the opportunity to clarify our motivation, methodology, and technical results.
> > > >
> > > >
> > > > If there are any further questions, comments, or suggestions, whether on technical points, presentation, or potential extensions, we would be happy to address them. Our goal is to ensure that the contribution is as clear, rigorous, and useful to the community as possible, and we value any additional input you might have.
> > > >
> > > >
> > > > Thank you again for your constructive engagement with our work.

---

### Official Review · Reviewer_wHsc · 2025-07-02

**Clarity:** 3
**Significance:** 2
**Originality:** 3
**Rating:** 4
**Confidence:** 3

**Summary:**

This paper aims to provide a solid theoretical foundation for Curriculum Learning (CL), an effective strategy in machine learning whose success is currently largely empirical and lacks sufficient theoretical understanding. To address this gap, the authors propose a theoretical framework based on biased Regularized Empirical Risk Minimization (RERM). The paper's core contribution is the introduction of a novel (r, a) condition to mathematically characterize a "good" curriculum.
Within this framework, the authors analyze the performance of CL across different learning tasks:
For convex learning tasks, they derive excess risk bounds and demonstrate that satisfying the (r, α) condition can effectively reduce sample complexity. This analysis is also extended to the widely used Stochastic Gradient Descent (SGD) algorithm.
The theoretical analysis is extended to non-convex tasks, providing generalization guarantees for the application of CL in modern scenarios such as deep learning.
Finally, the paper empirically validates its theoretical findings through experiments on synthetic and MNIST datasets.

**Questions:**

How robust is the model if early tasks are misleading or corrupted?
Will incorrect prior tasks lead to worse generalization, and how can this be mitigated?
Why are the datasets limited to synthetic and MNIST?

**Ethical Concerns:**

["NO or VERY MINOR ethics concerns only"]

**Final Justification:**

I have read the rebuttal by the authors and my concerns are addressed by it. I will increase my score.

**Limitations:**

(r, α)-condition is difficult to verify or enforce in practice. While theoretically elegant, it's unclear when or how this condition holds on real datasets.

**Quality:**

2

**Strengths And Weaknesses:**

Strengths:
1. The paper successfully identifies the disconnect between the practical application of CL and its theoretical underpinnings. Providing a theoretical basis for this widely used heuristic is a valuable and compelling research direction.
2. The proposed (r, α) condition is a central highlight of the work. It offers a novel, intuitive, and formal tool for defining what constitutes a "good" curriculum, providing a powerful instrument for analyzing CL's effectiveness.
3. The theoretical work in this paper is solid. The authors provide rigorous excess risk bounds for convex tasks and extend the analysis to the practical SGD optimizer, strengthening the connection between theory and practice.

Weaknesses:
1. While the theory is elegant, it provides limited insight into how to design a good curriculum in practice, especially for real datasets. There's no principled method to determine task order or difficulty.
2. Though the paper attempts to extend to nonconvex settings, the treatment is still mainly theoretical, and the methods like ERM over radius-constrained balls are not practical for large-scale deep learning.
3. Experiments are mainly conducted on synthetic data and MNIST, with relatively simple settings. The generalizability to complex domains (e.g., NLP, vision benchmarks) is not demonstrated.

---

> ### Author Rebuttal · Authors · 2025-07-29
>
> **(W1) Insight into Designing a Good Curriculum:**
>
> We respectfully disagree with the claim that our theory provides limited insight into curriculum design. While our primary goal is to formalize when curriculum learning helps, our framework also suggests how to design effective curricula. Specifically, under mild regularity assumptions (e.g., smooth loss, bounded hypothesis class), our results imply the following two practical principles (among others):
>
> -  Build curricula by expanding the hypothesis class incrementally, e.g., using norm-based constraints as in structural risk minimization.
> - Order training examples by increasing difficulty, such that the empirical distribution D_{t+1} is only a small shift from D_t, helping ensure the (r,α)-condition holds.
>
> These ideas are discussed (lines 269–284) and implemented in our adversarial robustness experiment, where the curriculum reflects increasing attack strength.
>
> Moreover, our theory provides tools for proving that a given curriculum will be effective, not just a post hoc justification, but a predictive framework for algorithmic design. We also note that curriculum learning differs from multitask learning: it does not assume a fixed pool of tasks to be permuted, but rather supports sequential design of tasks informed by learner progress, a perspective our framework explicitly supports.
>
>
>
> **(W2) Practicality of ERM over Radius-Constrained Balls in Non-Convex Settings:**
>
> We acknowledge the reviewer’s concern and agree that solving constrained ERM exactly is not practical in large-scale deep learning. However, the purpose of our non-convex analysis is to provide generalization guarantees for implicit approximations to this problem, such as those computed via SGD and backpropagation. While SGD does not solve ERM exactly, it is widely believed to approximate regularized ERM under certain stability or implicit bias conditions. Thus, theoretical analysis of constrained ERM remains meaningful, as it captures the idealized behavior that practical algorithms aim to approach. This is especially important since most generalization guarantees in deep learning are also based on such surrogate analyses. We will clarify this connection in the revision.
>
> **(W3) Scope and Simplicity of Experiments:**
>
> We appreciate the reviewer’s concern and agree that broader empirical validation is valuable. However, we emphasize that this is primarily a theoretical paper aimed at developing foundational understanding of curriculum learning, a widely used but poorly understood paradigm. Our analysis provides novel generalization bounds under the (r, α) condition, which we believe is a significant step forward in a largely under-theorized area.
>
>
> While the experimental section is limited in scope, it includes a nontrivial application to adversarial robustness, a realistic and practically relevant setting for both NLP/vision, demonstrating how our theory can guide curriculum design. The additional experiments on synthetic and MNIST data serve as proof-of-concept to validate key ideas in controlled settings. In particular, in Appendix D, we present an experiment on Noisy MNIST where the label space expands over time. We group digits into progressively larger categories to construct a curriculum with an increasing target label set, offering a simple empirical validation of our framework in such a setting. We view these results as complementary to the theory, not as the central contribution.
>
> **(Q1) Robustness to Misleading or Corrupted Early Tasks:**
>
> We appreciate this thoughtful question. It highlights an important distinction between curriculum learning and other learning paradigms. Unlike multitask learning or adversarial training, curriculum learning assumes a learner-driven or teacher-designed sequence of tasks. As such, the notion of robustness to arbitrary or corrupted early tasks is somewhat misplaced: if the curriculum is poorly constructed, one should not expect curriculum learning to help.
>
>
> Indeed, our framework makes this precise: if early tasks do not satisfy the (r, α) condition, then the curriculum may not improve generalization and may even hinder it. Our theory not only formalizes when a curriculum works, but also helps explain when and why it fails. In this way, it offers diagnostic insight into the success or failure of curriculum learning strategies.
>
>
> The goal of this paper is to move toward a formal definition of what makes a curriculum “good.” The (r, α)  condition provides this structure, and supports intuitive strategies such as gradually expanding the hypothesis class or presenting examples in order of increasing difficulty. These are natural analogs to pedagogical design (e.g., learning algebra before calculus), and our theory helps justify them rigorously.
>
>
> **(Q2) On Mitigating the Impact of Misleading Early Tasks:**
>
>
>
> Regarding mitigation, the key is not to fix the learner, but to design better curricula, just as we would not attempt to "mitigate" the failure of teaching algebra to kindergartners, but would instead reorder the curriculum to build foundational skills first. Curriculum learning depends on inductive bias; our framework offers a formal tool for verifying when such bias is appropriate.
>
> In short:
> - **Why are early tasks misleading?** Possibly because the curriculum was poorly constructed.
> - **How can this be mitigated?** By applying structure, such as the $(r, \alpha)$-condition, to ensure tasks are ordered so that learning is cumulative.
> - **Is this a flaw?** No—this is an inherent aspect of any learning system that leverages prior knowledge. There is no free lunch in machine learning: learning always requires assumptions. In curriculum learning, the key assumption is that the early tasks help prepare the learner for the final task. Our framework does not claim to discover an optimal curriculum automatically; rather, it aims to formalize what makes a curriculum effective, and to provide a theoretical foundation for analyzing or comparing curricula.
>
>
>
> We will clarify this perspective more directly in the revised version.
>
>
>
> **(Q3) Dataset Choices:**
>
> Since this is primarily a theory paper, we used synthetic data and MNIST to validate core ideas in controlled settings. These domains allow us to construct curricula that clearly satisfy or violate the (r, α) condition, making them ideal for illustrating our theoretical claims. We agree that extending to more complex datasets is a valuable direction for future work. Also, see response above labeled (W3).
>
>
>
>
> **(L1) Practical Verifiability of the (r, α) condition:**
>
> We agree that the (r, α) condition may be difficult to verify directly in practice. However, its value lies not in being easily checkable, but in serving as a guiding principle for curriculum design. Much like how educational curricula are designed to build progressively, ensuring that foundational concepts are mastered before introducing more advanced material, the (r, α) condition captures the idea that transitioning from task t to task t+1 should not require a complete re-learning of the problem. In this sense, it offers a formal lens through which to reason about inductive biases and curriculum structure, and can inform the development of heuristic or adaptive curriculum strategies in real-world settings. We will clarify this role in the final version.

---

> > ### Author Response · Authors · 2025-08-07
> > **Thank you!**
> >
> > We would like to thank you for your thoughtful review and for engaging deeply with our paper. We hope that our rebuttal helped clarify our motivation, theoretical contributions, and how the framework can inform curriculum design in practice.
> >
> >
> > As you consider your final assessment, we kindly ask that you take our responses into account, especially where we have addressed your concerns on practical insight, non-convex analysis, experiment scope, and robustness to misleading tasks.
> >
> >
> > If there are any remaining questions, comments, or suggestions, whether technical, methodological, or regarding presentation, we would be happy to provide additional clarifications. Our goal is to ensure the paper is as clear, rigorous, and useful to the community as possible, and we greatly value your input toward that end.
> >
> >
> > Thank you again for your constructive engagement with our work.

---

### Comment · Area_Chair_3RY5 · 2025-08-04
**Engage in discussion**

Dear reviewers,

Thank you for your thoughtful feedback! As the rebuttal comments are now available, we kindly encourage you to read the other reviews and the authors’ response carefully. If you have any follow-up questions, please raise them soon, so the authors can respond in a timely manner.

AC

---

### Decision · Program_Chairs · 2025-09-17

**Decision:**

Accept (poster)

**Comment:**

This paper received three borderline accept ratings and one strong accept rating. The reviewers found several weaknesses, which were addressed by the rebuttal comments. The paper is technically solid, providing a strong theoretical contribution that addresses an important topic, which justifies an accept decision on behalf of the AC.